# Dark microbiome and extremely low organics in Atacama fossil delta unveil Mars life detection limits

Armando Azua-Bustos [1,2] ✉, Alberto G. Fairén[1,3], Carlos González-Silva[4], Olga Prieto-Ballesteros[1], Daniel Carrizo [1], Laura Sánchez-García [1], Victor Parro[1], Miguel Ángel Fernández-Martínez[5], Cristina Escudero[1], Victoria Muñoz-Iglesias [1], Maite Fernández-Sampedro[1], Antonio Molina[1], Miriam García Villadangos [1], Mercedes Moreno-Paz [1], Jacek Wierzchos[6], Carmen Ascaso[6], Teresa Fornaro [7], John Robert Brucato [7], Giovanni Poggiali [7], Jose Antonio Manrique[8,9], Marco Veneranda [8], Guillermo López-Reyes [8], Aurelio Sanz-Arranz[8], Fernando Rull[8], Ann M. Ollila[10], Roger C. Wiens[10], Adriana Reyes-Newell[11], Samuel M. Clegg[10], Maëva Millan[12,13,14], Sarah Stewart Johnson [12,15], Ophélie McIntosh[7,15], Cyril Szopa [15], Caroline Freissinet [15], Yasuhito Sekine [16,17], Keisuke Fukushi [17], Koki Morida[18], Kosuke Inoue[18], Hiroshi Sakuma [19] & Elizabeth Rampe [20]

Identifying unequivocal signs of life on Mars is one of the most important objectives for sending missions to the red planet. Here we report Red Stone, a 163-100 My alluvial fan–fan delta that formed under arid conditions in the Atacama Desert, rich in hematite and mudstones containing clays such as vermiculite and smectites, and therefore geologically analogous to Mars. We show that Red Stone samples display an important number of microorganisms with an unusual high rate of phylogenetic indeterminacy, what we refer to as "dark microbiome", and a mix of biosignatures from extant and ancient microorganisms that can be barely detected with state-of-the-art laboratory equipment. Our analyses by testbed instruments that are on or will be sent to Mars unveil that although the mineralogy of Red Stone matches that detected by ground-based instruments on the red planet, similarly low levels of organics will be hard, if not impossible to detect in Martian rocks depending on the instrument and technique used. Our results stress the importance in returning samples to Earth for conclusively addressing whether life ever existed on Mars.

Past, current, and future missions to Mars are primarily motivated by the outstanding question of whether life ever existed on the red planet[1]. Landed missions like the Mars Exploration Rovers, Phoenix, and the active Mars Science Laboratory (MSL) and Mars2020 rovers were tasked with identifying habitable environments, and whether there are evidences for the requirements for life as we know it[2,3]. Liquid water is one of the main requirements, so many rovers have landed at sites with geomorphological evidence for ancient rivers and lakes and/or mineralogical evidence for liquid water, like clay minerals[4–6]. These spacecraft are equipped with various compositional instruments to

**Fig. 1 | Red Stone location and geological characteristics. A** Red Stone Location in the Atacama Desert (digital terrain model[86]). **B** Satellite image zooming in the surrounding region (Sentinel 2020 satellite data). **C** Paleogeographic reconstruction of the alluvial/delta system shown in **B**, according to the reported sedimentary record of the Caleta Coloso-El Way formation (modified from Flint, S. Clemmey, H. & Turner, P. The Lower Cretaceous Way Group of northern Chile: an alluvial fan–fan delta complex Sediment. Geology 46, 1-22 (1986)[25,87]). The black arrows in **C** show

the direction of the flow of the ancient river delta, which point of origin is now under the Pacific Ocean. The red point in **A**, **B** and **C** shows the location of the outcrop shown in **E**. **D** Stratigraphic column of the outcrop studied shown in panel E. Red arrows show sample collection points. **E** Close-up view of the studied outcrop. From top to bottom: UZ upper zone, U1 Unit 1, WI wall in sensor location, U2 Unit 2, WO wall out sensor location, LZ lower zone.

identify minerals and look for raw molecules required for life. Mass spectrometers on Viking, Phoenix, MSL, Mars2020 and the future ExoMars rover, for example, can detect organic molecules and the building blocks for life[7–10]. Although no robust evidences for organics in Martian soils were found by Viking or Phoenix measurements[11,12], both the Sample Analysis at Mars (SAM) instrument suite on MSL and the Scanning Habitable Environments with Raman and Luminescence for Organics and Chemicals (SHERLOC) instrument on Mars2020 have identified simple aliphatic and aromatic organic molecules (i.e., ~450 ppm in Yellowknife Bay mudstone at Gale crater[13,14]).

Results from Mars so far suggest that organics are not prevalent on its surface, but here we hypothesize that current instrument limitations[15] and the nature of organics in Martian rocks may also hinder our ability to find evidences of life on the red planet. In this work we test these limitations by closely inspecting Red Stone, a unique site located in the Atacama Desert, the driest[16–18], oldest desert on Earth[19–24], and a well-known Mars analog model[22].

## Results and discussion
### Site location and description
Red Stone is located South of the city of Antofagasta in the Quebrada del Boku (Boku gully) (Fig. 1A, B and Fig. 2), part of the upper Way Group, a sedimentary sequence of an alluvial fan–fan delta composed of the Coloso and Lombriz formations dating to the Early Cretaceous to Late Jurassic (Fig. 1C and Fig. 2B)[25]. The Way Group represents two distinct phases of basin evolution, recording the complete proximal to distal delta succession to the ulterior progressive marine incursion and delta deposition[26]. The base of the Lombriz formation shows intercalated red sandstones and mudstones with profuse perpendicular veins and indurated halite crusts (Fig. 1D, E and Fig. 3). Moving up

sections, there are units of cemented conglomerates, intercalated sandstones and mudstones, topped by weathered loose conglomerates (Fig. 1E and Fig. 3).

The secondary mineral assemblage of Red Stone sedimentary facies draws similarities to ancient Mars, suggesting comparable diagenetic histories. In addition to quartz and plagioclase, sandstones are composed of zeolite (analcime), calcite, gypsum, halite, chlorite, vermiculite, illite/muscovite and hematite (Fig. S1A). Hematite is a marker of oxidative aqueous alteration on Mars[27–29] and is present in Red Stone as well-developed crystals and thin mineral coatings (Fig. S1B, C), thus giving a characteristic red color to this site (Fig. 2A). The presence of halite and gypsum, crosscutting white veins and crusts in the lower Red Stone sandstone and mudstone facies, in addition to halite cementing the upper sandstone facies (Fig. 1D, E and Fig. 3), support arid conditions during the deposition of the sediments in this delta (halite and gypsum have been identified in ancient Martian surfaces from orbit and in situ[30–33]). Phyllosilicates are abundant in ancient Martian surfaces and are dominated by smectite and chlorite[34]. Similarly, Red Stone mudstones contain smectite (saponite and montmorillonite) and chlorite, along with illite and vermiculite (Fig. S1A and Fig. S1D). The presence of chlorites suggests low-temperature (70-90 °C) diagenetic conditions[35], similar to the temperatures experienced by sedimentary rocks in Gale crater investigated by MSL[36]. X-ray diffraction (XRD) patterns at different relative humidity (RH) show that that the peak position of chlorite slightly moves due to spreading of the interlayer distance, suggesting that chlorite may be absorbing small amounts of water vapor in response to changes in environmental relative humidity (RH) (Fig. S2), a finding of interest for the inspection of the potential habitability of this type of clay on Mars. Analcime cement, along with

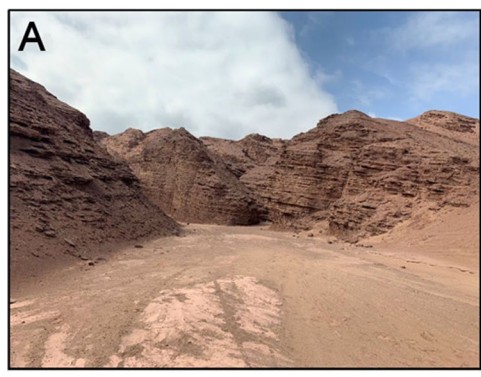

**Fig. 2 | Red Stone site characteristics. A** Panoramic view of the inspected site. **B** Geological map of the Caleta Coloso-El Way formation (SERNAGEOMIN, 2003. Mapa Geológico de Chile: versión digital[88]). JK1c (light green); transitional alluvial, fluvial and aeolic sedimentary sequences. Early Cretaceous to Late Jurassic (sandstone, limestone, lutite, conglomerate, siltstone). Ki1m (green); coastal marine sedimentary sequences. Early Cretaceous (sandstone, limestone, loam, calcarenite). M1c (light brown); alluvial fan sedimentary sequences. Miocene (sand, gravel, silt, ignimbrite). J3i (light purple); volcanic continental and marine Jurassic sequences (basalts, andesite, tuff, limestone).

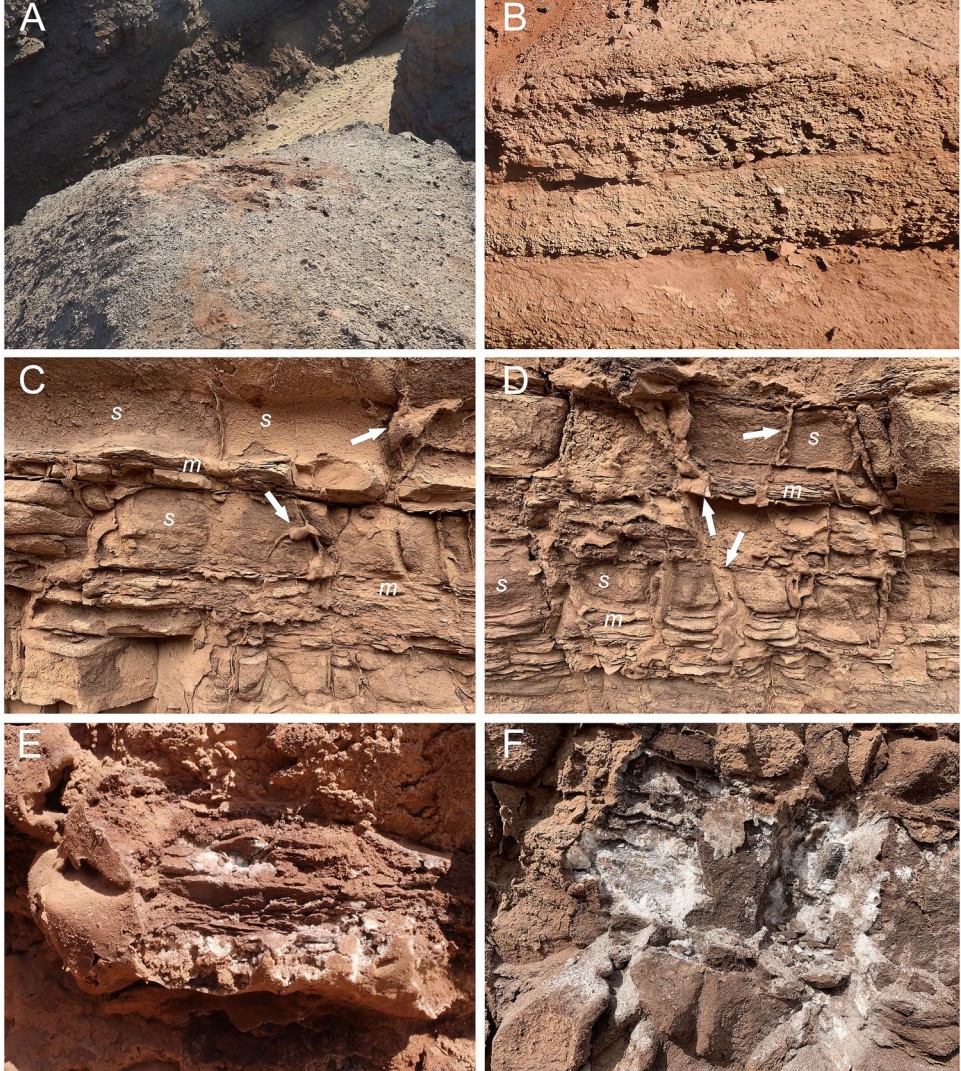

**Fig. 3 | Close-up views of the studied outcrop. A** Top weathered loose conglomerates. **B** Cemented conglomerates. **C, D** Show sandstones (s) with fine layered mudstones (m). White arrows point to halite/gypsum veins. **E** One of the evaporite veins after been broken, exposing the halite/gypsum inside them and the thin layer of hematite that covers them. **F** Fibrous halite crust parallel to the surface of the outcrop, only observable after been exposed.

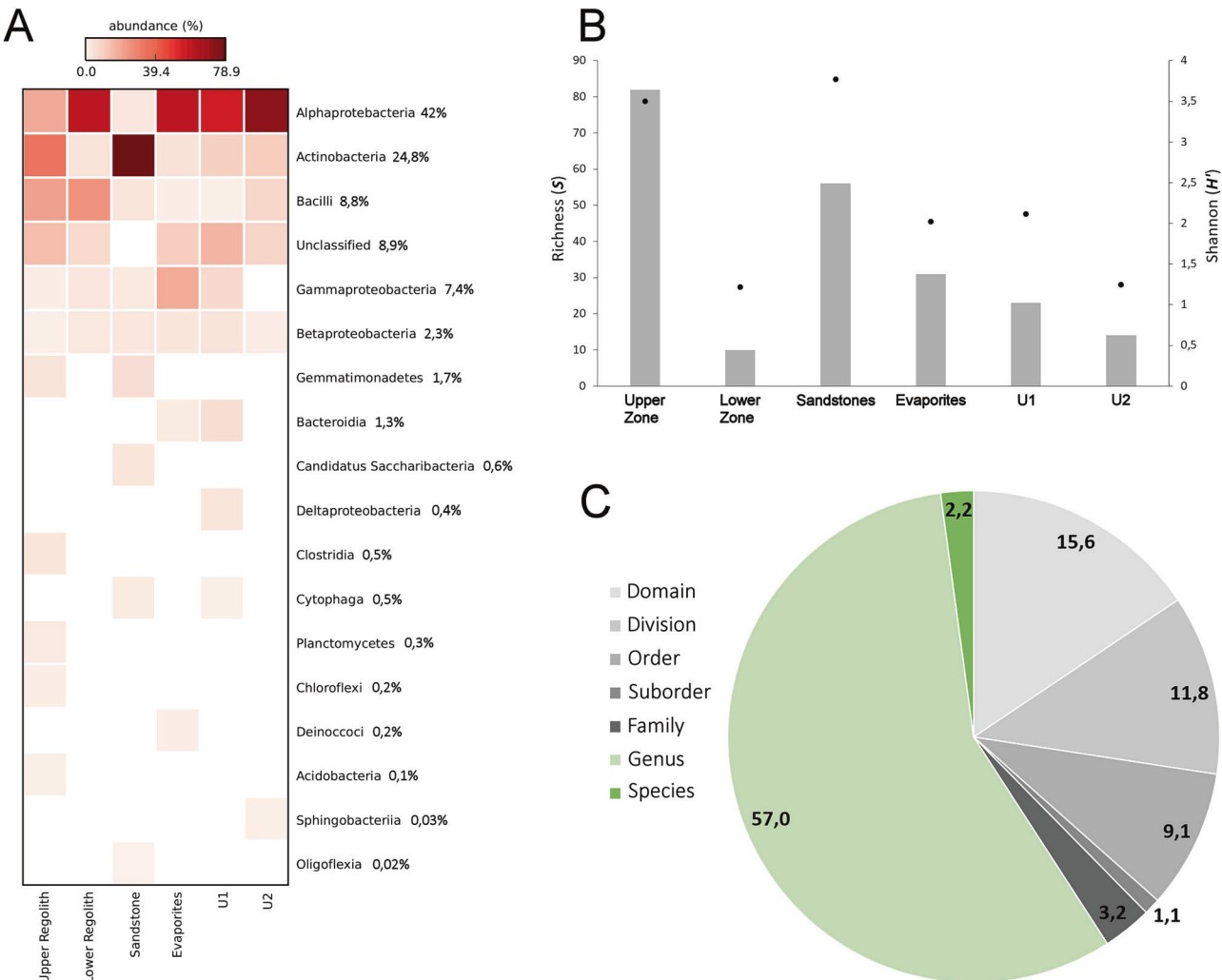

**Fig. 4 | Next-Generation Sequencing (NGS) characterization of Red Stone samples. A** Heatmap of major bacterial phyla determined by NGS. Numbers represent percentages of total sequences. The color intensity in each box indicates the relative percentage of each phyla. **B** Richness (S, bars) and Shannon diversity (H′, dots) indices for the analyzed prokaryotic communities. **C** Hierarchy classification of the bacterial sequences found by NGS. Gray scale colors identify higher hierarchy classifications.

the reported authigenic clays, calcite, and quartz[37], is typical of evaporating brines in closed basins, akin to lakes that formed in impact basins on Mars ~3-4 billion of years ago[38]. Weathering patterns using A-CN-K [$Al_2O_3$-(CaO + $Na_2O$) − $K_2O$] analysis[39,40] (Fig. S3A), along with A-C-N [$Al_2O_3$-(CaO + $Na_2O$)] and A-CNK-FM [$Al_2O_3$ − (CaO + $Na_2O$ + $K_2O$) − (FeOT + MgO)] geochemical analyses (Fig. S3B, C) are also indicative of diagenesis, and variations in the A-C-N in some of the samples also suggest other processes in action such as hydrothermalism.

In situ dual temperature/relative humidity loggers showed that the upper more exposed loose conglomerates displayed the highest RH values (Fig. S4), with up to 85.1% overnight and in the early morning hours, consistent with the regular exposure to fogs in this region as the main source of water for microbial life[41].

**Site microbial diversity and dark microbione**
Red Stone samples contain at most 1 µg of DNA per gram of soil, with 16 S rRNA Next Generation Sequencing (NGS) analyses showing that the highest number of species was among the Alphaproteobacteria and the Actinobacteria (Fig. 4A and Table S1). The highest diversity of Operational Taxonomical Units (OTUs) was found in the uppermost, weathered conglomerates (Fig. 4B), consistent with the highest availability of water. A correlation between the highest RH values and NGS

microbial diversity confirmed the high number of OTUs at the uppermost conglomerates (Fig. S5A), also unveiling a slightly higher number of OTUs at the zone where evaporites are located, in line with the role of these minerals in providing a source of water for microbial life due to water vapor absorption in response to changes in RH. Upper Zone OTUs are also found at the zone where the highest surface temperatures are recorded during the day time hours (Fig. S4 and Fig. S5B), suggesting that such species are at least thermotolerant.

A substantial fraction of the NGS 16 S rRNA sequences fell in the category "unclassified" (8.9%) (Fig. 4A), while 40.8% of the remaining sequences could not be assigned beyond higher taxa such as the order or domain (Fig. 4C), thus unveiling an unusual high degree of phylogenetic indeterminacy. The concept of "microbial dark matter" has been recently proposed to refer to the still unexplored portion of microbial diversity composed of uncharacterized microorganisms from known phyla and candidate phyla without cultivated representatives[42–44]. Here we instead propose the new concept of "dark microbiome" to refer to the community of microorganisms that can be detected with high throughput methods such as direct DNA sequencing (as is the case of Red Stone samples), but which phylogenetic identity still cannot be determined. Thus, the Red Stone dark microbiome may be composed by truly novel extant species not found anywhere else on Earth, but it may also be the case that such dark microbiome in fact

**Table 1 | Organic geochemistry of Red Stone samples with relative abundance of total organic carbon (TOC) (% dry weight), stable isotopic composition of the bulk organic carbon (‰), and concentration of lipid biomarkers (µg·g dw$^{-1}$)**

|  | Upper Zone | U1 | U2 | Sandstones | Mudstones | Evaporites spot1 | Evaporites spot2 | Lower Zone |
|---|---|---|---|---|---|---|---|---|
| TOC (% dw) | 0.10 | 0.02 | 0.02 | 0.02 | 0.10 | 0.10 | 0.11 | 0.06 |
| δ13C (‰) | −26.5 | −26.3 | −25.6 | −25.8 | −21.3 | −20.6 | −19.5 | −25.3 |
| *n*-alkanes (16-31) | 0.12 | 0.14 | 0.25 | 0.09 | 0.48 | 0.07 | 0.54 | 0.13 |
| pristane | 0.01 | n.d. | n.d. | n.d. | n.d. | n.d. | 0.01 | 0.01 |
| phytane | n.d. | n.d. | n.d. | n.d. | n.d. | n.d. | 0.01 | n.d. |
| heptadecene | 0.13 | 0.07 | n.d. | n.d. | n.d. | n.d. | n.d. | 0.19 |
| Hopanes | n.d. | n.d. | n.d. | n.d. | 0.06 | n.d. | n.d. | n.d. |
| br-alkanes[a] | 0.07 | 0.06 | 0.08 | 0.08 | 0.05 | 0.06 | 0.13 | 0.05 |
| *n*-fatty acids (14-20) | 1.85 | 0.95 | 1.35 | 1.47 | 1.92 | 1.75 | 2.42 | 0.59 |
| MUFA (16-18)[b] | 0.10 | 0.05 | 0.06 | 0.03 | 0.10 | 0.06 | 0.04 | 0.05 |
| *Iso*-C17:0[c] | n.d. | n.d. | n.d. | n.d. | n.d. | n.d. | 0.01 | n.d. |
| MMe-10-C16:0[d] | n.d. | n.d. | n.d. | n.d. | 0.28 | n.d. | n.d. | n.d. |
| Total sum of fatty acids[e] | 1.95 | 1.91 | 1.41 | 1.50 | 2.30 | 1.81 | 2.47 | 0.64 |
| *n*-fatty acids/ *n*-alkanes | 16 | 7 | 5 | 17 | 4 | 25 | 4 | 4 |

Lipid concentration was obtained by integrating the peak areas of each compound and applying the response factor of deuterated analogs (tetracosane-D$_{50}$ in the apolar fraction and myristic acid-D$_{27}$ in the acidic fraction) of known concentration added to the sample prior to extraction. The range of carbon numbers is indicated in brackets for the straight chain or *normal* (*n*) alkanes and fatty acids, as well as for the monounsaturated fatty acids (MUFA). n.d.: not detected,
[a]Sum of branched alkanes, in particular monomethyl alkanes of 16, 18 and 20 carbons,
[b]Sum of monounsaturated fatty acids from 16 to 18 carbons,
[c]Hexadecanoic acid, 15-methyl,
[d]10-Methyl hexadecanoic acid.
[e]Total sum of fatty acids, including *normal*, branched and unsaturated moieties.

represents the relict community of microbial species which used to inhabit the Red Stone delta in the distant past, of which no extant relatives are to be found in the existing sequence databases.

A culture-dependent approach allowed us to obtain only nineteen unique bacterial, and two fungal, isolates from the whole set of Red Stone samples (Fig. S6), with extremely low numbers of colony forming units (CFU) ($1.1 \times 10^1$ – $9.0 \times 10^1$, Table S2), and almost all of the bacterial species belonging to the Bacillaceae family of heterotrophic bacteria, in agreement with NGS findings. The majority of these species uniquely matched the mineralogy of the samples from where they were isolated (i.e., *Halobacillus* was found in evaporite samples), thus reflecting their ecological preferences. All the bacteria isolated from this site are known to use aeolian dust to move across the Atacama[45], suggesting that the coastal region in which Red Stone is located is the main point of origin of many of the species found inland.

**Red Stone Biosignatures**
Elemental analysis unveiled very low total organic carbon (TOC) contents coming from this unique assemblage of microorganisms, with a maximum of 0.11% dry weight, and no detectable nitrogen (Table 1). Gas chromatography–mass spectrometry (GC-MS) showed that the lipid fraction of this TOC was mainly composed of hydrocarbons and fatty acids (Fig. 5A). Stable carbon isotopic analysis showed δ$^{13}$C values from −19.5 to −26.5‰ (Fig. 5B), with evaporite samples being most enriched and having the highest TOC. Lipids characteristic of eukaryotic membranes, such as sterols, could not be detected by GC-MS. The presence of fatty acids with a methyl group at the penultimate position of the acid chain (i.e., *iso*−17:0) suggest bacterial inputs where sulfate-reducing bacteria may represent a substantial fraction[46]. The $^{13}$C enrichment observed in evaporites relative to other samples, along with the detection of monounsaturated fatty acids, isoprenoids (pristane and phytane), heptadecene, and a number of mid-chain monomethyl alkanes, suggest an origin in phototrophs like cyanobacteria[47–49]. However, since neither NGS, microscopic analyses (bright field and chlorophyll autofluorescence), nor culturing detected such species, these biosignatures may in fact represent the ancient

remains of the phototrophic members of the dark microbiome that inhabited the Red Stone alluvial fan before it was lithified. This hypothesis is also coherent with the analysis of the mudstone samples, which also revealed the presence of a series of hopanes from C27 to C31 carbons (Fig. 5A). Hopanes are the most recalcitrant molecular fossils of hopanoids, steroid-like lipids from the isoprenoid family produced primarily by bacteria[50], again a potential biosignature of the ancient inhabitants of the Red Stone delta.

The ratio of the sum of *n*-fatty acids over the sum of *n*-alkanes may provide a measure of the biomass freshness in a given environment. As oxygen-functional groups such as carboxyl tend to decompose over time, while more resistant hydrocarbons accumulate (i.e., alkanes), relatively high *n*-fatty acids/*n*-alkanes ratios can be interpreted as a sign of fresher biomass[51]. Samples from the upper part of the stratigraphy and evaporites showed the highest ratios (≥16; Table 1), suggesting that these samples contain the fresher/more recent biomass, consistent with the highest microbial diversity determined by NGS and the highest availability of water in these samples.

Raman spectroscopy using a 532 nm source, similar to the Raman on the Mars 2020 rover's SuperCam instrument suite, confirmed the presence of the minerals detected by XRD (Fig. S7), but could not find any lipids signatures (Fig. 5C–F), unveiling the critical proper choice of Raman parameters such as laser source and spot size in the detection of different types of organics when concentrations are extremely low.

SYBR Green DNA staining was successful for a single sample of the lower zone, which unveiled a few coccoid and bacillar cells (Fig. S8). However, Catalyzed Reporter Deposition-Fluorescence in situ Hybridization (CARD-FISH) was able to detect extremely low levels of bacteria and archaea (Figs. S9, S10) in all samples, with the highest counts (only $6 \times 10^5$ cells per gram of sample) observed in the top conglomerates samples, again consistent with these samples having the highest biodiversity and water availability.

**Mars testbed instrument results**
Given that several of the techniques used to first characterize Red Stone samples gave evidences of life at the limit of detection, it was of

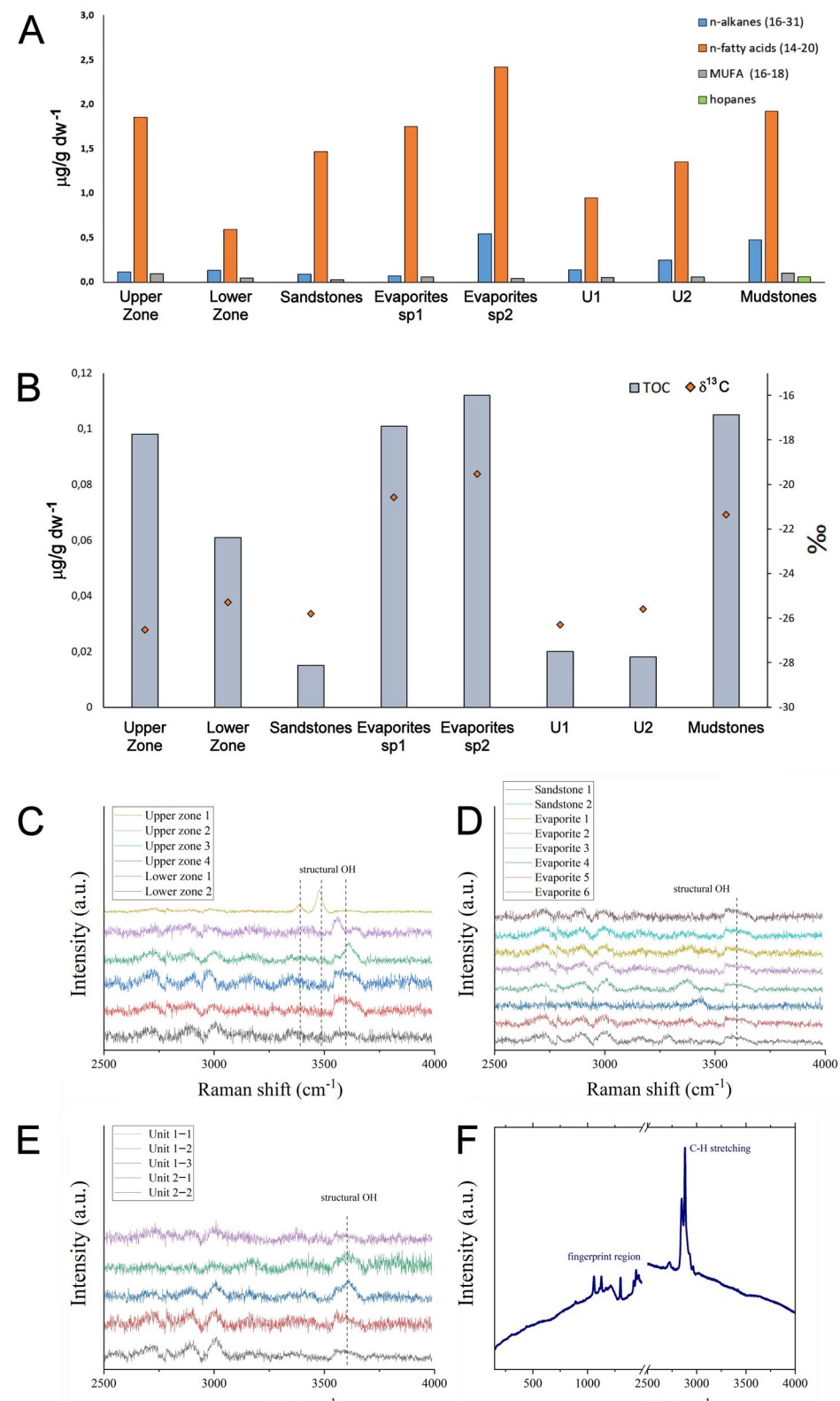

**Fig. 5 | Lipid Biosignature analyses of Red Stone samples. A** Lipid content, including normal (i.e. straight chain) alkanes, normal fatty acids, and mono-unsaturated fatty acids (MUFA) identified by gas-chromatography mass spectro-metry (GC-MS). Carbon numbers ranges are indicated in brackets. **B** Total organic carbon (TOC, % dw) and stable carbon isotopic composition (δ13C, ‰) values found. **C**, **D** and **E** show the samples analyzed through Raman spectroscopy, while (**F**) shows a positive control in which a lipid was deposited on an amorphous silica-rich substrate (panel F modified with permission from Carrizo et al.[89]).

interest to inspect what a suite of testbed instruments on and to be sent to Mars would detect in Red Stone samples, as current and future missions are specifically investigating similar clay rich fan/fan-deltas[52,53].

While techniques such as DRIFTS (Diffuse Reflectance Infrared Fourier Transform Spectroscopy), RLS (Raman laser spectrometer ExoMars simulator), and LIBS (Laser Induced Breakdown Spectroscopy onboard the Curiosity and Perseverance rovers) closely matched the mineralogical composition of the site (Figs. S11–S13, Table S3 and Table S4), the detection of organics proved to be much more challenging.

DRIFTS detection of organics was barely possible in the NIR region (Fig. S11A), as stronger bands from vibrations in mineral lattices obscured potential bands from organics. Only a single peak at 1.36 μm in the Red Stone sample spectra could be explained by non-fundamental bands of organics. In contrast, many (although very weak) bands attributable to organics were observed in the mid-infrared (MIR) spectral region (Fig. S11B; assignments in Table S5).

SAM-like pyrolysis[54] of Red Stone samples detected a number of different organics (Fig. S14), including aromatics, oxygen-containing aromatics, chloroaromatics, and sulfur-bearing aromatics. Specific organics, such as C12-C35 straight-chain alkanes in UZ, U1 and U2 samples were detected in low relative abundance compared to the other organics (Fig. S14) and identified by their mass spectra and retention times using a C10-C40 alkanes standard. The fact that alkanes were detected at the limit of detection of the commercial instrument used indicate that they might not be detectable using the SAM flight model, which has a limit of detection about ten times lower compared to the commercial version used here (~ppbv). However, SAM-like derivatization wet chemistry experiments using MTBSTFA-DMF (see methods) identified polar molecules and derivatized $C_7$-$C_{20}$ $n$-carboxylic acids in all samples, unveiling proline as the only amino acid detected (only in the top conglomerate samples, Fig. S15), consistent with the active metabolism of bacteria in this particular sample. The detection of dicarboxylic and unsaturated fatty acids, along with the finding of $C_{16}$ and $C_{18}$ fatty acids and proline, suggest that these organics would be detected by the SAM instrument on Mars. However, this would be only possible if these molecules are abundant, and subjected to instrumental variables that could play an additional role in their detection, such as the temperature constraints of the instrument (column, trap, transfer lines, etc.), the time limit of the flight-analysis, the desorption capability of the trap(s), etc.

When Red Stone samples were analyzed via flash pyrolysis with MOMA[55] (the Mars Organic Molecule Analyzer on board the Rosalind Franklin ExoMars rover) testbed instrument, no organics were detected (Fig. S16). However, when these samples were derivatized with MTBSTFA-DMF (direct derivatization and extraction prior to derivatization), a few peaks were identified, but only in evaporite samples; aliphatic carboxylic acid species derivatized through silylation of their labile group −OH (Fig. S17). These results not only unveiled that most Red Stone samples contained organic levels below MOMA detection limits, but also, the critical importance of analog field samples such as Red Stone to validate the next generation of flight instruments.

Red Stone samples were also analyzed with SOLID-LDChip, the Signs of Life Detector[56–58], a TRL5 instrument designed to search for molecular biosignatures on other planets by using the LDChip (Life Detector Chip), a multiplex fluorescence sandwich immunoassay. Similar to other testbed analyses, SOLID detections were just over the detection limit in Red Stone samples (Fig. S18), but still finding evidences of heterotrophic bacteria and interestingly, cyanobacteria, consistent with other biosignatures coming from the extant and relict microorganisms using other techniques. The detection by SOLID of ancient remnants of cyanobacteria is of particular interest, as it confirms that the Red Stone river delta had enough water to support

photosynthesis millions of years ago, but not anymore as the Atacama got drier in time (similar to the case of Mars).

## Relevance of red stone data for Mars life detection
Altogether, Red Stone offers a unique collection of geological and microbiological characteristics, providing an exceptional analog model site to study the formation and the extant and past habitability of an ancient alluvial fan/fan delta that formed under arid conditions in a Mars analog setting[59]. The analyses of Red Stone samples using rover testbeds unveil the relevance of wet chemistry derivatization experiments for the detection of organics in mass spectrometers, despite this technique was not always able to identify all types of organics. In addition, the SOLID immunoassay approach showed to be a promising technique for detecting evidences for microbial life, rather than just its building blocks, although Martian microorganisms present in concentrations lower than those in Red Stone may still not be detectable. The limited or non-detection by rover testbed instruments of a number of biosignatures of the unique extant and extinct microbes in Red Stone samples also highlight the critical importance of a Mars Sample Return mission, so samples can be thoroughly studied for signs of life in laboratories on Earth[60].

## Methods
### Site sampling
Main field recognition and sampling of the Red Stone outcrop took place on August, 2019, although subsequent site sampling also took place in February, May, August and October of 2021. After surveying the surroundings, the most continuous exposure was selected to take advantage of the best section traverse of the stratigraphic record. We took reference and close-up photos before and after collecting material, enough to sample unaffected by weathering and the necessary amount for geochemical analysis. The sampling was methodically logged including field observations and their depth with a tape measure as perpendicular as possible to the bedding plane, and stored in zip bags and glass tubes with aluminum foil lids for biosignatures analyses with appropriate labeling. Some fragile formations (coatings and salt layering) were stored additionally in falcon tubes, to be able to observe and analyze properly later in the lab. We took 17 samples representative of the different horizons along the stratigraphic sequence (some required additional later sampling as aforetold) that was represented in the form of a graphic log plot using Strater v5 (Golden Software).

### X-Ray diffraction
Powder X-Ray Diffraction of bulk samples was performed using a Bruker D8 Eco Advance with Cu Kα radiation and Lynxeye XE-T linear detector. Samples were scanned between 5° (2θ) and 60° (2θ) using a step size of 0.05° (2θ) and a count time of 1 s. The phase identification was performed by comparing the measured diffraction pattern with patterns of the PDF Database with the DIFFRAC.EVA software (Bruker AXS). Afterwards, the clay fraction was obtained by decantation according to Stokes' law. The determination of clay involved treatment: air dried, solvation with ethylene glycol, and heating at 350 and 500 °C during 2 h. Then Samples were scanned with a 0.02° (2θ) step size over the range 2-30°(2θ) with a 1 s collection time at each step.

### X-Ray diffraction analysis under different relative humidity
The oriented specimens of the rock samples were analyzed by XRD (Ultima IV) attached with a humidity control device in National Institute for Materials Science, Japan. The measurements were made with monochromatic CuKα radiation at 40 kV and 30 mA. The oriented specimens were dried in a vacuum-oven at 100 °C for 15 h before the measurements. The dried specimens were placed in the sample chamber. The XRD patterns of the samples were measured under the relative humidity from 1% to 60%.

## Geochemical analyses

To graphically examine weathering trends, we included some of the most widely used ternary diagrams that allow for the graphical interpretation of the proportional chemical changes. The A-CN-K, A-C-N and A-CNK-FM ternary diagrams, ploted using Grapher v13 (Golden Software).

## Environmental data

Temperature and relative humidity were measured in the field using dual iButton temperature/Humidity micro loggers (Maxim Integrated, San Jose, CA, USA) as previously done[18,23], and set to take data every 15 min for 2 months.

## SYBR green bacterial counts

Red stone samples were stored at room temperature and then enumerated by direct counts in the lab. As cell visualization is less efficient when minerals mask cells by adsorbing traditional dyes, we used SYBR Green I (Molecular Probes, Eugene, USA) instead of DAPI (4′,6-Diamidino-2-Phenylindole, Dihydrochloride). Briefly, 2 g of each sample were suspended in 10 mL of 0.01 M tetrasodium pyrophosphate and gently sonicated in a Branson ultrasonic cleaner (Danbury, USA) for 30 min in ice cooled water. After 10 s of sedimentation, 2 mL of the supernatant were taken and placed directly on black Isopore polycarbonate membrane filter (0.2 μm pore size, Millipore, Massachusetts, USA). Membranes were then incubated with 1.5 mL of SYBR Green I (50 μg/L) during 15 min in the dark. Membranes were then washed with 10 mL of distilled water and left in dark for air drying. One drop of immersion oil was deposited on glass slide, then the filter and another drop of oil was deposited on the filter surface and covered with a glass cover. Bacteria were counted immediately using Zeiss AxioImager M.2 fluorescence microscope (Carl Zeiss, Jena, Germany) and a Plan-Apo 63x/1.4 Zeiss oil-immersion objective. Filter set for eGFP (Zeiss Filter Set 38; Excitation/Emission: 450–490/500–550 nm) was used for SBI stained bacteria visualization. Bacteria were counted by selecting random fields of view, with 20 fields per filter analyzed. Five filters were counted per sample, for a total of 10 filters and 200 fields observed.

## CARD-FISH

Samples for Catalized Reporter Deposition Fluorescence in situ Hybridization (CARD-FISH) were fixed with 4% formaldehyde in PBS and stored at 4° until further analysis. Microorganisms were detached from mineral particles by sonication and filtered in 0.22 mm black membranes (Millipore, Germany) in aseptic conditions as described in ref. [61]. CARD-FISH experiments, counterstaining and their respective autofluorescence and unspecifically binding controls were performed as previously described in detail[62]. CARD-FISH were realized using EUB338 I-III mix probes[63,64] and ARC915[65]. Furthermore, CARD-FISH with NON338[66] were realized as additional hybridization control. Samples were visualized and imaged as already reported[61].

## DNA extraction

Red Stone samples were stored at room temperature and DNA extracted from them in the lab using the DNeasy PowerSoil Pro Kit (Quiagen, Düsseldorf, Germany) according to the manufacturer instruction and as previously performed[18,23,41].

## Illumina NGS-based 16 S and 18 S rRNA Sequencing

DNA concentrations were quantified by Picogreen. Then, variable input of DNA and variable number of cycles were used in a first PCR with Q5® Hot Start High-Fidelity DNA Polymerase (New England Biolabs) in the presence of:

100 nM primers for 16 S amplification (5'-ACACTGACGACAT GGTTCTACACCTACGGGNGGCWGCAG-3' and 5'- TACGGTAGCAGAGA CTTGGTCTGACTACHVGGGTATCTAATCC-3', these primers amplify

the V3-V4 region of 16 S), 200 nM primers for 18 S amplification (5'-A CACTGACGACATGGTTCTACAGCCAGCAVCYGCGGTAAY-3' and 5'-TA CGGTAGCAGAGACTTGGTCTCCGTCAATTHCTTYAART-3'), 200 nM primers for Archaea amplification (5'-ACACTGACGACATGGTTCTACA CGGRAAACTGGGGATAAT-3' and 5'-TACGGTAGCAGAGACTTGGT CTTRTTACCGCGGCGGCTGBCA-3'), 200 nM primers in the case of ITS (5'-ACACTGACGACATGGTTCTACATCCTCCGCTTATTGATATGC-3' and 5'-TACGGTAGCAGAGACTTGGTCTGTGAATCATCGAATCTTTGAA-3') and 200 nM primers for Cyanobacteria (5'-ACACTGACGACAT GGTTCTACAGGGGAATYTTCCGCAATGGG-3' as Forward, and 5'-TACG GTAGCAGAGACTTGGTCTGACTACTGGGGTATCTAATCCCATT-3' or 5'-TACGGTAGCAGAGACTTGGTCTGACTACAGGGGTATCTAATCCCTT T-3' as Reverse). After the first PCR, a second PCR of 12 or 14 cycles was performed with Q5® Hot Start High-Fidelity DNA Polymerase (New England Biolabs) in the presence of 400 nM of primers (5'-AATGAT ACGGCGACCACCGAGATCTACACTGACGACATGGTTCTACA-3' and 5'-CAAGCAGAAGACGGCATACGAGAT-[10 nucleotides barcode]-TACGG TAGCAGAGACTTGGTCT-3') of the Access Array Barcode Library for Illumina Sequencers (Fluidigm).

The finally obtained amplicons were validated and quantified by Bioanalyzer and an equimolecular pool was purified by agarosa gel electrophoresis and titrated by quantitative PCR using the "Kapa-SYBR FAST qPCR kit forLightCycler480" and a reference standard for quantification. The pool of amplicons were denatured prior to be seeded on a flowcell at a density of 10pM, where clusters were formed and sequenced using a "MiSeq Reagent Kit v3", in a 2 × 300 pair-end sequencing run on a MiSeq sequencer. We note here that no archaea or eukaryotic sequences (except a few well-known fungal contaminants) were found, thus only 16 S data is presented in the main report. All raw sequence data were deposited at the NCBI Sequence Read Archive (SRA, http://www.ncbi.nlm.nih.gov/sra) under accession number PRJNA908755.

## Biodiversity analyses

Raw sequences were processed in MOTHUR software v.1.43.0[67], using a custom script based upon MiSeq SOP[68], as previously employed before[69–73]. Microbial OTU richness (S) and Shannon diversity index (H′) were calculated with R language for statistical computing (R Core Team, 2019), by using "vegan" package v.2.5-6[74]. For the better visualization of the bacterial community composition, a heatmap was constructed by means of STAMP software v.2.1.3[75].

## Isolation of soil microorganisms

In the lab, samples were stored at room temperature and then aseptically inoculated in Petri dishes containing agar and either Luria–Bertani Broth (Sigma-Aldrich, Missouri, USA), Marine Media or modified Czapek Dox growth media (CondaLab, Torrejón de Ardoz, Spain). This was done by directly sprinkling a total of 100 mg per soil sample into the Petri dishes containing the aforementioned media, incubating these plates at 25 °C. In most cases colonies arising from soil particles were evident 2 weeks after of inoculation. All colonies were then re-cultivated in the media from they were first isolated, to obtain enough biomass for subsequent DNA extraction and storage.

## DNA extraction from isolates

Samples were stored at room temperature and DNA extracted from them in the lab using the DNeasy UltraClean Microbial Kit (Quiagen, Düsseldorf, Germany) according to the manufacturer instruction and as previously performed[18,21,41].

## Isolate identification

As previously performed[18,21,41], 16 S rRNA genes of bacterial isolates were amplified in the lab using the GoTaq Green Master Mix (Promega, Wisconsin, USA) and the primers 341 f (5′CCT ACG GGNGGC WGC

AG3′) and 785r (5′GAC TAC HVGGG TAT CTA ATC C′). PCR conditions used were: 95 °C for 5 min, and 25 cycles of (95 °C for 40 s, 55 C for 2 min, 72 °C for 1 min) followed by 72 °C for 7 min. 18 S rRNA of fungal isolates were amplified in the lab using the GoTaq Green Master Mix (Promega, Wisconsin, USA) and the primers F566 (5′CAG CAG CCG CGG TAA TTCC3′) and R1200 (5′CCC GTG TTG AGT CAA ATT AAG C3′). PCR conditions used were: 95 °C for 15 min, and 35 cycles of (95 °C for 45 s, 60 °C for 45 s, 72 °C for 1 min) followed by 72 °C for 10 min. The resultant reactions were visualized in a 2% agarose TAE gel at 50 V. The automated sequencing of the resulting PCR products was conducted by Macrogen DNA Sequencing Inc. (Seoul, Korea). Sequences were checked for quality using the BioEdit software (Ibis Therapeutics, Carlsbad, USA) and end-trimmed before using the Megablast option for highly similar sequences of the BLASTN algorithm against the National Centre for Biotechnology Information nonredundant database (www.ncbi.nlm.nih.gov) to search for the closest species of each of the isolates obtained.

Phylogenetic analysis of 16 S rRNA and 18 S rRNA isolate gene sequences was performed by aligning sequences by multiple sequence comparison by log-expectation, analyzed with jModelTest and then by Phylip NJ (bootstrap 10,000), all tools of the freely available Bosque phylogenetic analysis software (version1.7.152)134. All isolate genes sequences have been deposited at the NIH genetic sequence database (https://www.ncbi.nlm.nih.gov/genbank/) both for 16 S rRNA (OP955639 to OP955657) and 18 S rRNA sequences (OP954719 and OP962462).

### Biosignature analyses (GC-MS)
Lyophilized subsamples of soils (10 to 20 g), were spiked with internal standards of $n$-alkanes (tetracosane-$D_{50}$) and $n$-alkanoic acids (myristic acid-$D_{27}$) and then extracted with a mixture of dichloromethane and methanol (DCM:/MeOH, 3:1, v/v) by ultrasound sonication (3 ×10 min cycles). The total lipid extract (TLE) was concentrated to ~0.5 ml using rotary evaporation and then digested overnight at room temperature in a mixture of methanolic potassium (6% w/w), further separated into neutral and acidic fractions. The neutral lipid fraction was obtained by extracting the methanolic potassium mixture with 30 ml of $n$-hexane (Hx) three times, rotavaporating and recovering it with ~1 ml of Hx:DCM (9:1, v/v). The acidic lipid fraction was obtained by adding HCl to the remaining methanolic potassium mixture and extracting it with 30 ml of Hx (three times), then it was concentrated using rotary evaporation and collected with DCM. Further separation of the neutral fraction into non-polar (hydrocarbons) and polar sub-fractions was done by eluting the concentrated neutral fraction (~1 ml of Hx:DCM) on an alumina column using ~0.5 g of $Al_2O_3$ powder in a precombusted Pasteur pipet with 4.5 ml of Hx:DCM (9:1, v/v) and 3 ml of DCM:methanol (1:1, v/v), respectively. Both non-polar and acidic fractions were analyzed by gas chromatography mass spectrometry (GC-MS), by direct injection on Hx, the former, or by injection after derivatization with $BF_3$ in MeOH to form fatty acid methyl esters (FAME). GC-MS analysis was made using a 6850 GC system coupled to a 5975 VL MSD with a triple axis detector (Agilent Technologies) operating with electron ionization at 70 eV and scanning from $m/z$ 50 to 650. 1 μl of analytes were injected and separated on a HP-5MS column (30 m x 0.25 mm i.d. x 0.25 μm film thickness) with He as a carrier gas at a constant flow of 1.1 ml·min$^{-1}$. For analyzing the non-polar fraction, the oven temperature was programmed to gradually increase from 50 °C to 130 °C at 20 °C·min$^{-1}$ and then to 300 °C at 6 °C·min$^{-1}$ (held 20 min). For analyzing the acidic fraction, the oven temperature was programmed from 70 °C to 130 °C at 20 °C·min$^{-1}$ and then to 300 °C at 10 °C·min$^{-1}$ (held 10 min). The injector temperature was set at 290 °C, the transfer line was at 300 °C and the MS source at 240 °C. Compounds identification was based on the comparison of mass spectra with reference materials, and their quantification on the use of external calibration curves of $n$-alkanes ($C_{10}$ to $C_{40}$) and fatty acid methyl esters

(FAME; $C_8$ to $C_{24}$). All chemicals and standards were supplied by Sigma Aldrich (San Luis, Missouri, USA). The recovery of the internal standards averaged 71 ± 14%.

### Stable isotopes analysis
Stable isotope composition of organic carbon ($δ^{13}C$) was measured on the bulk soil samples using isotope-ratio mass spectrometry (IRMS), following the USGS method[76]. Briefly, all samples were homogenized by grinding with a mortar and pestle. They were then decarbonated (soils) with HCl (3 N) and, after 24 h of equilibration, they were adjusted to neutral pH with ultrapure water. The samples were dried in an oven (50 °C) until constant weight. Ratios of $δ^{13}C$ and $δ^{15}N$ were measured in a MAT 253 IRMS (Thermo Fisher Scientific, Waltham, Massachusetts, USA) and reported in the standard per mil notation (‰). Three certified standards were used (USGS41, IAEA-600 and USGS40) with an analytical precision of 0.1 ‰. The content total organic carbon (TOC %) was measured with an elemental analyzer (HT Flash, Thermo Fisher Scientific, Waltham Massachusetts, USA) during measurement of the stable isotopes.

### Raman spectroscopy
Raman spectra were performed exciting the sample with a non-polarized Nd: YAG solid state laser of 532 nm wavelength. After focusing onto a monochromator (Horiba Jobin Yvon HRi550), with a diffraction grating of 1200 grooves/mm, the scattered light was detected with a Charge Coupled Device (CCD), 1024×256 pixels, cooled to 203 K for thermal-noise reduction. The spectrometer is connected by fiber optics to a B&W Tek microscope with a 50x objective that allows a spot size on the sample of 42 μm (Microbeam S. A., Spain). The spectral resolution, with slit width of 200 μm, results better than 5 cm-1. Raman spectra were taken in a laser power range of 20 – 150 mW, 10–100 s of integration times and 3 accumulations.

### DRIFTS analyses
Infrared characterization of Red Stone samples was performed at INAF-Astrophysical Observatory of Arcetri (Florence, Italy) using a single beam double pendulum interferometer VERTEX 70 v (Bruker), equipped with a Praying Mantis™ Diffuse Reflection Accessory (Harrick DRIFT) in order to carry out Diffuse Reflectance Infrared Fourier Transform Spectroscopy (DRIFTS) analysis. Specifically, we acquired reflectance spectra in the 400-8000 cm$^{-1}$ spectral region, using a resolution of 4 cm$^{-1}$, a Globar source, a KBr beamsplitter, a DTGS detector, and in the 8000-20000 cm$^{-1}$ spectral region, using a resolution of 8 cm$^{-1}$, a Tungsten source, a CaF$_2$ beamsplitter and a InGaAs detector, in order to cover the entire spectral range relevant to space flight instruments such as SuperCam (spectral range 1.3-2.6 μm + 0.4–0.85 μm) on board NASA *Mars 2020* rover mission, Ma_MISS (spectral range 0.5-2.3 μm), ISEM (spectral range 1.1–3.3 μm) and MicrOmega (spectral range 0.95-3.65 μm + 0.5–0.9 μm) on board ESA-Roscosmos *ExoMars 2022* rover mission.

### Raman laser spectroscopy analyses
Red Stone samples were crushed on an agate mortar and sieved using a set of sieves of different pore size, in order to obtain particle sizes of equal or less to 250 μ FT Raman was performed using a Bruker RFS 100 instrument with an excitation source of 1064 nm and a spot of 100 microns. Two spectra from two different points were acquired using a total of 1024 scans per spectra. Micro Raman analyses at 633 nm. For this end an excitation source of 633 nm, and a NIKON microscope as focusing/collection optics were used, giving variable spot sizes of the analyses, depending of the objective, going down to 25 micron spots. High performance spectroscopy was performed with a Holospec 1.8i and a Raman head from Kaiser Optical Systems. For operator guided analyses, four different spectra were analyzed for major components, along the focus on individual grains. For the RLS simulator, the

automatic acquisition over a total of one hundred 50 micron spots were acquired, with an automated system described in ref. [77].

## Laser induced breakdown spectroscopy analyses

Five observations of 30 laser shots each were made on the samples with the lab replica of the ChemCam instrument. Samples were situated 1.6 meters from the instrument, contained in a chamber surrounded by 7 Torr of CO2 to imitate the Mars atmosphere. LIBS data were pre-processed using the methods described in[78] to remove non-laser background and continuum, and to calibrate for wavelength. TRLS (time-resolved luminescence spectroscopy) observations were made with a lab replica of the SuperCam instrument.

## SAM-like pyrolisis

Flight-like pyrolysis experiment were performed using a Frontier Laboratories 3030D multi-shot pyrolyzer from the Quantum Analytics company. Samples were powdered and aliquots (~20 mg) were deposited into the stainless-steel pyrolysis cup. Samples were heated from 40 to 850 °C at 35 °C/min which reproduced the rate of the pyrolysis ramp used on the Sample Analysis at Mars instrument (SAM) onboard the Curiosity rover[79]. Gas-chromatography mass spectrometry (GC-MS) experiments were conducted on a trace 1310 GC coupled to an ISQ quadrupole mass spectrometer (both from the Thermo Fisher company). The GC was equipped with a Restek capillary MXT-5 column (30 m long x 0.25 mm ID bonded with a 0.25 μm thick stationary phase) similar to one used by the SAM and allowing to analyze and separate a wide range of molecules ($-C_5$-$C_{30}$). The GC ramp was initially heated at 35 °C and held for 5 min followed by a 5 °C/min ramp up to 300 °C, held for 2 min. The carrier gas was helium and set at 1.2 ml/min. The samples were analyzed in splitless mode and the temperatures of the injector, transfer line and ion source were all set at 300 °C. The MS was set to scan ranges of masses between m/z 40 and 535 amu. During the duration of the pyrolysis, the volatiles were trapped at the column inlet by a glass liquid nitrogen cold trap located inside the GC oven, right after the GC injector. By the time the pyrolysis ended, the cold trap stopped and heated, so the GC analysis can begin. Blanks were performed between each simple analysis to prevent and control potential contaminations. MTBSTFA-DMF is one of the two wet chemistry reagents present on the SAM instrument[54]. MTBSTFA derivatization is a non-selective silylation technique that facilitates the extraction and detection of polar molecules (e.g., amino acids, fatty acids) that cannot be directly analyzed by pyrolysis-GCMS and was demonstrated to work on Mars to detect polar molecules[80]. The SAM derivatization experiments were mimicked in the laboratory by depositing ~20 mg of solid sample in the pyrolysis cup, which itself was placed into a 2 ml open vial. The samples were spiked with 40 nmol of DL-Fluorovaline used as an internal standard to control the derivatization efficiency after analysis. The samples were then dried for ~48 h at 40 °C to limit interference from the MTBSTFA reagent with the water adsorbed on the sample. Before the pyrolysis analysis, 20 uL of MTBSTFA-DMF were deposited into the sample cup and heated at 75 °C for 15 min after the vial was sealed. In addition, to mimic the first SAM heating step, this temperature has previously been optimized and demonstrated to be the most efficient to derivatized standard polar molecules. After heating, the cup was quickly loaded into the pyrolyzer to minimize MTBSTFA solvent evaporation. The sample was then heated up from 75 °C up to 850 °C in SAM-like conditions using the SAM heating rate of 35 °C/min. The chromatographic column was initially heated at 40 °C (no hold) followed by a first 10 °C/min ramp up to a temperature of 80 °C, followed by a second temperature ramp at 5 °C/min up to a final temperature of 310 °C held for 5 min to bake out the column before the next analysis. As for the pyrolysis experiment, the carrier gas used was helium and set at a 1.2 ml.min$^{-1}$ flow rate with a split flow of 75 ml/min to limit the saturation of the reagents. The MS ion source and transfer line were both set at 300 °C and ions produced by the 70 eV electron ionization source were scanned with mass-to-charge ratios (m/z) comprised between 40 and 535 with a scan time of 0.2 s from 9 min (solvent delay) to the end of the run.

## MOMA flash pyrolysis and sample derivatization

Pyrolysis: All instrumental procedures were performed using a Frontier Laboratories 3030D multi-shot pyrolyzer (FrontierLab) mounted on the split/splitless injector of a Trace Ultra gas chromatograph coupled to an ISQ quadrupole mass spectrometer (Thermo Fisher).

20 mg of each sample were weighted in an organic-cleaned steel cup (microscale precision: 0.01 mg) and place at the top of the pyrolyzer (cold). The cup was then lowered into the pyrolyzer's oven heated at either 400 °C or 600 °C and stayed for 30 s at this temperature. The gaseous content of the sample was directly transferred to a MOMA-like MXT-5 column (20 m long; 0.25 mm internal diameter; 0.25 μm stationary phase thickness) (Restek) for chromatographic separation. The injector was set at 250 °C and the GC temperature program started at 40 °C followed by a ramp of 10 °C. min$^{-1}$ to reach a 300 °C final temperature, and stayed for 10 min at 300 °C. As for the mass spectrometer, the ion source and transfer line temperature were heated to 300 °C, and the electron beam of the ion source set at 70 eV to scan the ion mass to charge ratio (m/z) from 10 to 600.

Derivatization: All samples were derivatized with a silylating reagent present in the MOMA instrumental suite, N,N-methyl-tert-butyl-dimethylsilyltrifluoroacetamide (MTBSTFA). N,N-dimethyl-formamide (DMF) was added to MTBSTFA (1:4) as a solvent and to help promote the derivatization reaction[81]. Two methods of derivatization were conducted: a direct derivatization method and extraction prior to derivatization.

a.  Direct derivatization; 50 mg of each sample were transferred into a glass vial (microscale precision: 0.01 mg). 100 μL of derivatization reagent and 1 μL of internal standard (methyl laurate at $10^{-2}$ M in ethyl acetate) were added. The solution was then heated at 75 °C for 15 min. 0.5 μL of the supernatant were manually injected in the GCMS instrument.

b.  Extraction method; Because the organic matter present in the samples was found to be either absent or of very low abundance with the pyrolysis experiment, we proceeded to an extraction prior to derivatization in order to concentrate the organic content. 50 mg of the sample were transferred into a glass vial (microscale precision: 0.01 mg). 250 μl of a (1:1) solution of water and methanol were added to the sample in order to extract water soluble compounds such as amino acids as well as other organics that have a higher solubility in organic solvents such as fatty acids. The screwed caps were additionally sealed with parafilm to prevent water leaks and the vials were put into an ultrasonic bath (Branson 2200) for 2 h. The liquid solution was then transferred into an Eppendorf tube and centrifuged to separate the phases. The tube was heated at 75 °C under a nitrogen flow until the liquid phase had evaporated. 50 μL of derivatization agent and 1 μL of internal standard (same as previously) were added. The sample was heated at 75 °C for 15 min and 0.5 μL of the supernatant was injected manually with a microsyringe to the GC. The potential presence of chemical contaminants from the Eppendorf tubes through heating was assessed by control GC-MS and no contamination was detected. Two replicas were done for each sample for both method to insure the repeatability of our experiments. All instrumental procedures were conducted on a Trace 1300 gas chromatograph coupled to a TSQ 9000 triple quadrupole mass spectrometer. We used a standard RTX-5 column (30 m long; 0.25 mm internal diameter; 0.25 μm stationary phase thickness) from Restek. The temperature program and mass spectrometer parameters were the same as for the pyrolysis experiments.

## SOLID-life detector chip

A 200 antibody-microarray-based immunosensor (LDChip or Life Detector Chip), the core of the SOLID (Signs of Life Detector) instrument developed for planetary exploration, targets a wide variety and molecular sizes of microbial biomarkers, mostly polymers[82–84]. The IgG fraction of polyclonal antibodies was printed in triplicate spot pattern microarray on microscope glass slides so than 9 complete LDChip can be set per slide as described previously[85]. The complete list of the antibodies on LDChip can be found in ref. [86]. Solid ground samples were processed following a similar procedure as the automatic one perfumed by SOLID instrument. Briefly, 0.5 g were resuspended in 2 mL of lysis/incubation tris-buffered saline with Twin20 double reinforced (TBSTRR) buffer (0.4 MTris-ClH, pH 8, 0.3 M NaCl, 0.1% Tween 20), ultrasonicated for $3 \times 1$ min pulses and filtered through 20 microns to remove coarse particulate matter[86]. Then, 50 uL of each crude extract were incubated for 8 h at 4 °C with LDChip in each corresponding chamber of the MultiArray Analysis Module (MAAM)[71] that simulate SOLID Sample Analysis Unit (SPU). Blank control only contained buffer. After washing away with TBSTRR buffer, all chambers were incubated with 50 uL of a mixture of fluorescently labeled antibodies (0.5-1 ug/mL each) for 12 h at 4 °C as tracers for the immunoreactions. After washing away and dry, the chips were scanned for fluorescence, the image processed, quantified and analyzed as described elsewhere[85,86].

## Data availability

Data and materials availability: all data and samples are freely available upon request to AAB. All NGS raw sequence data were deposited at the NCBI Sequence Read Archive (SRA, http://www.ncbi.nlm.nih.gov/sra) under accession number PRJNA908755. All isolate genes sequences have been deposited at the NIH genetic sequence database (https://www.ncbi.nlm.nih.gov/genbank/) both for 16 S rRNA (OP955639 to OP955657) and 18 S rRNA sequences (OP954719 and OP962462). Source data are provided with this paper.

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

## Acknowledgements

The research leading to these results is a contribution from the Project "MarsFirstWater", funded by the European Research Council, Consolidator Grant no 818602 to AGF, and by the Human Frontiers Science Program grant n° RGY0066/2018 to A.A.-B. Additional funding provided was provided by MINECO grant PID2019-107442RB-C32 (O.P.-B. and A.M.), Grants-in-Aid for Scientific Research from the Japan Society for Promotion of Science grant numbers 17H06458 and 21H04515 (K.F.), grant numbers 17H06456, 17H06458, 20H00195, and 21H04515 (K.F. and Y.S.), Consejería de Educación e Investigación, Comunidad Autónoma de Madrid/European Social Fund program (MAFM), grant n° ESP2017-87690-C3-3-R (DC), Ramón y Cajal grant n° RYC2018-023943-I (L.S.-G.), AEI grant MDM-2017-0737 and MCIN/AEI grant PID2019-107442RB-C32 (V.M.-I.), MCIU/AEI (Spain) and FEDER (UE) grant n° PGC2018-094076-B-I00 (J.W. and C.A.), Italian Space Agency agreement 2017-48-H.O (T.F., J.R.B. and G.P.), the Ministry of Science of Spain grant PID2019-107442RB-C31 (J.A.M., M.V., G.L.R., A.A. and F.R.), María Zambrano' excellence grant program (CA3/RSUE/2021-00405), funded by the Spanish Ministry of Universities (MFM), NASA Mars Exploration Program contracts NNH13ZDA018O, NNH15AZ24I, NNH13ZDA018O and LANL Laboratory Directed Research and Development (LDRD) funding XX5V (A.M.O, R.C.W., A.R. and S.M.C.), NASA-GSFC grant NNX17AJ68G (M.M. and S.S.J.), NES focused on Sample Analysis at Mars of the Mars Science Laboratory mission, and Mars Organic Molecules Analyzer of the Exomars 2022 mission (O.M., C.S., and C.F.), and grants RTI2018-094368-B-I00 and MDM-2017-0737 Unidad de Excelencia "Maria de Maeztu"- Centro de Astrobiología (INTA-CSIC) by the Spanish Ministry of Science and Innovation/State Agency of Research MCIN/AEI/ 10.13039/501100011033 and by "ERDF A way of making Europe" (C.E., M.G.V., M.M.-P., and V.P.). R.C.W. thanks Dot Delapp for performing pre-processing of the LIBS data. The authors also thanks USGS earth explorer for providing the satellite data of Sentinel 2020.

## Author contributions

Conceptualization: A.A.-B., A.G.F., C.G.-S. Methodology: A.A.-B., A.G.F. Investigation: A.A.-B., C.G.-S., O.P.-B., D.C., L.S.-G., V.P., M.A.F.-M., C.E., V.M.-I., M.F.-S., A.M., M.G.V., M.M.-P., J.W., C.A., T.F., J.R.B., G.P., J.A.M., M.V., G.L.-R., A.S.-A., F.R., A.M.O., R.C.W., A.R.-N., S.M.C., M.M., S.S.J., O.M., C.S., C.F., Y.S., K.F., K.M., K.I., H.S. Visualization: A.A.-B., A.G.F., C.G.S., O.P.B., D.C., L.S.-G., V.P., M.A.F-M., C.E., V.M-I., M.F.-S., A.M., M.G.V., M.M.-P., J.W., C.A., T.F., J.R.B., G.P., J.A.M., M.V., G.L.-R., A.S.-A., F.R., A.M.O., R.C.W., A.R.-N., S.M.C., M.M., S.S.J., O.M., C.S., C.F., Y.S., K.F., K.M., K.I., H.S., E.R. Validation: A.A.-B., A.G.F., C.G.-S., O.P.-B., D.C., L.S.-G., V.P., M.A.F.-M., C.E., V. M.-I., M. F.-S., A.M., M.G.V., M.M.-P., J.W., C.A., T.F., J.R.B., G.P., J.A.M., M.V., G.L.-R., A.S.-A., F.R., A.M.O., R.C.W., A.R.-N., S.M.C., M.M., S.S.J., O.M., C.S., C.F., Y.S., K.F., K.M., K.I., H.S., E.R. Supervision: AGF. Writing-original draft: A.A.-B. Writing—review and editing: A.A.-B., A.G.F., C.GS., O.P.-B., D.C., L.S.-G., V.P., M.A.F-M., C.E., V.M-I., M.F-S., A.M., M.G.V., M.M.-P., J.W., C.A., T.F., J.R.B., G.P., J.A.M., M.V., G.L.-R., A.S.-A., F.R., A.M.O., R.C.W., A.R-N., S.M.C., M.M., S.S.J., O.M., C.S., C.F., Y.S., K.F., K.M., K.I., H.S., E.R.

## Competing interests

The authors declare no competing interests.

## Additional information

[1]Centro de Astrobiología (CAB) (CSIC-INTA), 28850 Madrid, Spain. [2]Instituto de Ciencias Biomédicas, Facultad de Ciencias de la Salud, Universidad Autónoma de Chile, Santiago, Chile. [3]Department of Astronomy, Cornell University, Ithaca 14853 NY, USA. [4]Facultad de Ciencias, Universidad de Tarapacá, Arica, Chile. [5]Department of Ecology, Universidad Autónoma de Madrid, 28049 Madrid, Spain. [6]Museo Nacional de Ciencias Naturales (CSIC), 28006 Madrid, Spain. [7]INAF-Astrophysical Observatory of Arcetri, Florence, Italy. [8]Universidad de Valladolid, Valladolid, Spain. [9]Institut de Recherche en Astrophysique et Planétologie (IRAP), Toulouse, France. [10]Purdue University, Earth, Atmospheric, and Planetary Sciences, West Lafayette, USA. [11]Southwest Sciences, Inc. 1570 Pacheco St. Ste. E11, Santa Fe, NM 87505, USA. [12]Department of Biology, Georgetown University, Washington, DC 20057, USA. [13]NASA Goddard Space Flight Center, Solar System Exploration Division, Greenbelt, MD 20771, USA. [14]LATMOS/IPSL, UVSQ Université Paris-Saclay, Sorbonne Université, CNRS, 11 Bd d'Alembert, 78280 Guyancourt, France. [15]Science, Technology, and International Affairs Program, Georgetown University, Washington, DC 20057, USA. [16]Earth-Life Science Institute (ELSI), Tokyo Institute of Technology, Tokyo, Japan. [17]Institute of Nature and Environmental Technology, Kanazawa University, Kanazawa, Japan. [18]Division of Natural System, Kanazawa University, Kanazawa, Japan. [19]National Institute for Materials Science, Tsukuba, Japan. [20]Astromaterials Research and Exploration Science Division, NASA Johnson Space Center, Houston, TX, USA. ✉e-mail: aazua@cab.inta-csic.es

