## [Peer Review File · Nature Communications]

Dark microbiome and extremely low organics in Atacama fossil delta unveil Mars life detection limitsReviewer #1 (Remarks to the Author):

The key message of this paper is that evidence of ancient life on Mars is unlikely to be found with the flight instruments that are currently on Mars or are planned for upcoming Mars missions. They conclude that sample return will be needed to have the best chance of identifying that evidence. The authors collected samples from a deltaic deposit that is visually very similar to the deltaic deposit currently being studied and sampled on Mars by the Perseverance rover. The sediments in the deposit are at least 100MY old, and based on the history and nature of the deposits, biological activity must have been ongoing during the deposition. Samples were analyzed by numerous state-of-the-art laboratory instruments, and with these methods subtle biosignatures were observed but only with GCMS type organic analyzers and only when derivatization methods were incorporated. The paper argues that the flight versions of the instruments would not have detected these signatures. This is an important conclusion, and the paper should be published. In fact, it is concerning that even the state-of-the-art laboratory instruments did not find strong signals of life in terrestrial samples only 100MY old and currently experiencing habitable conditions and active microbiota. Compared to Martian conditions even in the distant past, these samples are in an oasis for life. If we can barely detect biosignatures of life in these samples, we should not be confident that we will find them in similar samples returned from Mars that were laid down billions of years ago. And we should be careful to interpret negative results for biosignatures, if this occurs in returned samples from Mars, as an indication that life did not occur on Mars. While this point is not stated, it is an important lesson from this suite of analyses.

Another important and surprising result is that the Signs of Life Detector (SOLID) instrument did detect evidence of life. I would expect that the instrument is detecting signatures of the active biology present today. However, the other methods (DNA extraction and staining) did not detect evidence of cyanobacteria as detected by SOLID. This instrument currently has no path to a flight mission and has only been considered for flight missions to search for extant life. If SOLID is detecting a signature of cyanobacteria dating from when the sediments were deposited, that would be an important endorsement of this instrumental method in the search for evidence of ancient biosignatures. This point is not argued for or against by the authors, but I recommend something more be said about it.

Below are answers to specific questions recommended by the journal to be addressed in the review.

- Will the work be of significance to the field and related fields? How does it compare to the established literature? If the work is not original, please provide relevant references.

I think this work, especially published in Nature, will be significant in the field by making the wide community aware of the difficulty of finding clear biosignatures of ancient life from the methods traditionally used in spaceflight instruments. The methods and instruments used in the analysis of these samples are extensive and strongly support the paper's conclusions. The methods section is very detailed and meets or exceeds common standards in the field. In fact, there are few examples where any Mars analog sample has been subjected to all the different instruments and methods used in this study.

- Does the work support the conclusions and claims, or is additional evidence needed?

As there is extensive evidence presented, no additional evidence is needed. I did not find any flaws in the data analysis but admit that I do not have extensive experience using any of the methods by which the samples were analyzed.

- Is there enough detail provided in the methods for the work to be reproduced?

The methods section has extensive detail and should allow reproduction of any of the results (but see my above admission).

Clarity and context:

The manuscript is well written and clear. I found a few minor issues with the language use and have flagged these in the pdf files from the article and the supplementary information.

References.

I noted one paper that surely should be cited but was not, and that may be because this paper came out after the manuscript was prepared. I note in the line by line comments below where it should be referenced.

Specific comments on the manuscript.

Parts of the text in figure 1C are cut off on the left side. What are the black arrows in 1C? They point away from the shoreline. Surely the flow must be towards the sea, not away from it.

Line 82. The PNAS paper Stern et al 2022 is reporting ~450 ppm organic carbon. This result should be mentioned here. The reference is Jennifer C. Stern et al. Organic carbon concentrations in 3.5-billion-year-old lacustrine mudstones of Mars. PNAS 2022. <https://doi.org/10.1073/pnas.2201139119>

Line 108. The reference list stops at 30 in the article but the rest of them are in the supplemental file. Is this a Nature protocol?

Line 118. Add the word "with" after the word "along"

Line 121. Same as above

Lines 165-169. These lines suggest that there is a detectable ancient biosignature from the organic analysis. But the conclusion of the paper is that there is almost no such evidence even with these state of the art instruments. This seems a contradiction.

Line 187-188. Lipids were not detected here by Raman, but were by GCMS, suggesting Raman on Mars will not detect lipids deposited on early Mars. So what is Sherlock Raman detecting?

Line 238. This emphasizes that analysis of analog field samples with flight instruments provide important information about the meaning of their results on Martian samples. If these instruments can't find organics in terrestrial samples collected near the ocean, why should they be expected to find them on Mars from rocks laid down billions of years ago?

Line 244. SOLID detected signatures of cyanobacteria but previously it was stated that signatures of cyanobacteria were not observed in DNA analysis or in the organic analysis (lines 160-176) and that this implied that cyanobacteria signatures must not be modern. Does this imply that what SOLID is seeing is antigens of cyanobacteria laid down in the ancient sediments? If so, that would be a very important conclusion.

In their review of the first version of this manuscript, reviewer #1 added some comments to the manuscript file. These comments were forwarded to the authors, who replied as included in this Peer Review File.

Reviewer #2 (Remarks to the Author):

This paper highlight the microbial community inhabiting soil samples collected from Red Stone, Atacama Desert and its environmental conditions using the combination of multidisciplinary techniques. Soil parameters of Red Stones were also determined and compared with those detected by ground based instruments. It is an advanced attempt that provide thorough data for the Mars mission where each techniques and parameters were selected and conducted rigorously that will give good impact in science field.

The introduction of 'dark microbiome' term is up to date and relevant that can serve as the extension of 'microbial dark matter'. This is the major contribution by the authors which is in line with the advancement of NGS platforms and bioinformatic analysis approaches that contribute towards more explicit exploration in microbial taxonomy and microbial ecology field.

From the methodology provided, authors mentioned the use of 4 different types of primers in NGS namely 16S, 18S, archaea and cyanobacteria but only 16S results were presented.

- Please provide the data obtained from other primers**
- It is suggested that more analysis can be done on the data of different type of microbes and comparison can be made between them.**
- Multivariate analysis can be conducted to observe the relationship between microbiome data with environmental factors.**

Performing culture dependent technique provide a clearer view of the microbes in Red Stone as extended future studies can be carried out to discover their taxonomic classification and potential in various field. However, data revealed that only few species were isolated:

- Are those the only 26 isolates obtained in this study? Or there are more isolates but have not been identified based on 16SrRNA gene sequences? How was the colonies selected?**
- Please amend the methodology section so that it can be reproduced. This can be done by providing more details on isolation technique (incubation temperature, duration, how the colonies where selected from isolation plates, dilution factor and so on)**
- Conduct phylogenetic analysis on the 16S sequence of each isolates to get the accurate result of their evolutionary relationship and the taxonomic group that they belong to. BLAST result only provide indication of the closest hit from their database.**

Suggestions:

- A more optimized selective isolation technique to be applied in future as commented in the manuscript.**
- Apart from those data, the number of microbes in each soil samples can be estimated using cfu/g**

As the term 'dark microbiome' is used in the title that shows the importance of this part of study, it is vital for authors to utilize the data obtained and conduct more comprehensive analysis on them to improve this manuscript.

The other experiment such as SYBE green and CARD-FISH analyses contribute to the information of microbial life detected in the soil samples. In addition, the outcome of mineral and environmental data is making this study complete. It would be good if some data can be combined in one or more analysis (multivariate, biological modelling etc) to produce more absolute outcome of this multidisciplinary study.

I am not in the position to comment of other part of this paper as my expertise is in microbiology and microbial ecology field. However, I am positive that other reviewers will contribute their thoughtful comments that will help in making this paper a good one.

Thank you

In their review of the first version of this manuscript, reviewer #2 added some comments to the manuscript file. These comments were forwarded to the authors, who replied as included in this Peer Review File

Reviewer #3 (Remarks to the Author):

I was asked to review the organic geochemistry aspects of this manuscript which I have done.

On the organic geochemical side I found that it lacked detail and with the lack of chromatograms from the standard GC-MS analysis makes it difficult to assess. I felt that detailed organic geochemical descriptions are needed in the Supp Data section so this can be used as a benchmark for the SAM and MOMA-like data that is produced.

I was not convinced that organics were detected by Raman, NIR or MIR this needs to be addressed.

Peaks are mislabelled or misidentified in S14

All figures need improving with legends and peak labels etc.

On the organic geochemistry part of the manuscript I do not feel that it is up to the required standard currently for this journal.

Specific points/comments are below

Line 159: GCMS should be GC-MS.

Line 164: Iso fatty acids are common in numerous bacteria not just sulfate reducers

Line 168: How about other common cyanobacteria markers such as hopenes?

Line 173: Which hopane isomers/epimers some are produced by bacteria. Were hopenes detected such as diploptene which is common in cyanobacteria.

Line 177-178: Also dependent on the source not just a measure of metabolic activity

Line 179: Depends on the diagenetic conditions need to be anoxic otherwise the functionalised components just get further oxidised.

Lines 181-3: Over simplification as stated above could be down to different source and differing diagenetic/redox conditions

Line 184: What is the detection limit of Raman (it is generally an insensitive technique for organic detection) is it not well below the already stated TOC?

Line 201: What is the detection limit for organics by DRIFTS is it not above the already stated levels. Not convinced by the NIR or MIR traces that there are any peaks that can be unambiguously identified as being due to C-H stretching/overtone. NIR main C-H overtone is at 1700 nm. The main C-H peaks in MIR should be $\sim 2900\text{ cm}^{-1}$ but not highlighted if present. Approx MIR detection limits for orgC are $\sim 0.1\%$ under ideal conditions which these are not (e.g. DTGS detector etc.)

Figure S11: X-axis needs to be reversed and it is also usual to plot the Y-axis as $\log(1/R)$ instead of reflectance

Figure S14 has mis-label/identified compounds; C16 fatty acid would not elute next to the C16 alkane but just before n-C20. Why is only the m/z 57+71 mass chromatogram shown and not the TIC with the compounds mentioned in the main text (line 216). Peak labels need to be described in the legend

Line 214: No mention of the alkenes if pyrolysis is occurring the alkenes should be evident, if they are not is thermal desorption a more dominant process due to the relatively slow heating rate? No mention of how the mineral assemblage would affect the products seen in the pyrolysate, which can have a significant affect. Was no contamination detected (all samples are contaminated just to different extents)? Were any standards run to check the pyrolysis system was working correctly?

Figure S15: Are the peak labels alkanes or fatty acids? If they are fatty acids surely C16 should be the largest peak? The trace should bare some relationship to the straight pyrolysis runs which it does not appear to do. How will the minerals affect the derivatisation, release of bound water (as can be seen in S16) will both damper the

reaction and can make the derivatised products unstable.

Line 232: MOMA heating at 400 C is below pyrolysis temperatures and is thermal extraction

Table 1: The units are $\mu\text{g g}^{-1}$ dry weight as not as written in legend. No mention of where the values are derived for the concentrations

Figure S17: Why are the products not labelled?

Supp data

Biosignature analysis (GC-MS): Where are the chromatograms for the more traditional wet chemistry organic geochemical techniques. The recoveries are rather low

REVIEWER COMMENTS AND RESPONSES

Reviewer #1 (Remarks to the Author):

The key message of this paper is that evidence of ancient life on Mars is unlikely to be found with the flight instruments that are currently on Mars or are planned for upcoming Mars missions. They conclude that sample return will be needed to have the best chance of identifying that evidence. The authors collected samples from a deltaic deposit that is visually very similar to the deltaic deposit currently being studied and sampled on Mars by the Perseverance rover. The sediments in the deposit are at least 100MY old, and based on the history and nature of the deposits, biological activity must have been ongoing during the deposition. Samples were analyzed by numerous state-of-the-art laboratory instruments, and with these methods subtle biosignatures were observed but only with GCMS type organic analyzers and only when derivatization methods were incorporated.

The paper argues that the flight versions of the instruments would not have detected these signatures. This is an important conclusion, and the paper should be published. In fact, it is concerning that even the state-of-the-art laboratory instruments did not find strong signals of life in terrestrial samples only 100MY old and currently experiencing habitable conditions and active microbiota. Compared to Martian conditions even in the distant past, these samples are in an oasis for life. If we can barely detect biosignatures of life in these samples, we should not be confident that we will find them in similar samples returned from Mars that were laid down billions of years ago. And we should be careful to interpret negative results for biosignatures, if this occurs in returned samples from Mars, as an indication that life did not occur on Mars. While this point is not stated, it is an important lesson from this suite of analyses.

Another important and surprising result is that the Signs of Life Detector (SOLID) instrument did detect evidence of life. I would expect that the instrument is detecting signatures of the active biology present today. However, the other methods (DNA extraction and staining) did not detect evidence of cyanobacteria as detected by SOLID. This instrument currently has no path to a flight mission and has only been considered for flight missions to search for extant life. If SOLID is detecting a signature of cyanobacteria dating from when the sediments were deposited, that would be an important endorsement of this instrumental method in the search for evidence of ancient biosignatures. This point is not argued for or against by the authors, but I recommend something more be said about it.

Below are answers to specific questions recommended by the journal to be addressed in the review.

• Will the work be of significance to the field and related fields? How does it compare to the established literature? If the work is not original, please provide relevant references.

I think this work, especially published in Nature, will be significant in the field by making the wide community aware of the difficulty of finding clear biosignatures of ancient life from the methods traditionally used in spaceflight instruments. The methods and instruments used in the analysis of these samples are extensive and strongly support the paper's conclusions. The methods section is very detailed and meets or exceeds common standards in the field. In fact, there are few examples where any Mars analog sample has been subjected to all the different instruments and methods used in this study.

• **Does the work support the conclusions and claims, or is additional evidence needed?**
As there is extensive evidence presented, no additional evidence is needed. I did not find any flaws in the data analysis but admit that I do not have extensive experience using any of the methods by which the samples were analyzed.

• **Is there enough detail provided in the methods for the work to be reproduced?**
The methods section has extensive detail and should allow reproduction of any of the results (but see my above admission).

Clarity and context:

The manuscript is well written and clear. I found a few minor issues with the language use and have flagged these in the pdf files from the article and the supplementary information. References. I noted one paper that surely should be cited but was not, and that may be because this paper came out after the manuscript was prepared. I note in the line by line comments below where it should be referenced.

Our responses to specific comments on the manuscript by Reviewer #1

- **Parts of the text in figure 1C are cut off on the left side.**

You are right! We did not notice that, this has been corrected in the new Figure 1 (line 435 of the revised manuscript).

- **What are the black arrows in 1C? They point away from the shoreline. Surely the flow must be towards the sea, not away from it.**

This is a very pertinent question! The arrows in panel 1C show the direction of the flow of the ancient river delta. They point into the continent and not away from it, as previous reports show that the point of origin of the river delta is now under the Pacific Ocean. To clarify this, the meaning of the black arrows has now been added to the legend of Figure 1 as "*The black arrows in C show the direction of the flow of the ancient river delta, which point of origin is now under the Pacific Ocean*" (line 440 of the revised manuscript).

- **Line 82. The PNAS paper Stern et al 2022 is reporting ~450 ppm organic carbon. This result should be mentioned here. The reference is Jennifer C. Stern et al. Organic carbon concentrations in 3.5-billion-year-old lacustrine mudstones of Mars. PNAS 2022. <https://doi.org/10.1073/pnas.2201139119>**

We agree. As suggested, a brief comment on this reference, and the reference itself, has been added in line 82 of the revised manuscript: "*...instrument on Mars2020 have identified simple aliphatic and aromatic organic molecules (i.e., ~450 ppm in Yellowknife Bay mudstone at Gale crater)*¹³⁻¹⁵"

- **Line 108. The reference list stops at 30 in the article but the rest of them are in the supplemental file. Is this a Nature protocol?**

Yes, we based the structure of our manuscript based on a number of different Nature Communication articles. Nevertheless, we will ask the help of the editor on this.

- **Line 118. Add the word "with" after the word "along"**

Done!, now it is stated that "*Analcime cement, along with the reported authigenic clays...*" (Line 118 of the revised manuscript).

- Line 121. Same as above

Also done!: “...along with A-C-N and A-CNK-FM geochemical analyses...” (line 121 of the revised manuscript).

- Lines 165-169. These lines suggest that there is a detectable ancient biosignature from the organic analysis. But the conclusion of the paper is that there is almost no such evidence even with these state of the art instruments. This seems a contradiction.

This is also a very pertinent comment. That is why we say in line 55 “barely detected with state-of-the-art laboratory equipment”. In other words, organics were hardly detected with ground-based instruments (such as ¹³C enrichment in this case) (with a maximum TOC of 0.11 %; line 164). However, when testbed instruments on and to be sent to Mars were used, the organics detected were even less or nil. Details of this are better explained throughout lines 204 - 259 of the revised version of the manuscript.

- Line 187-188. Lipids were not detected here by Raman, but were by GCMS, suggesting Raman on Mars will not detect lipids deposited on early Mars. So what is Sherlock Raman detecting?

Again, this is a very pertinent question. This is because the lasers sources used in Raman spectroscopy range from the UV into the near-infrared and beyond. Thus, the Raman apparatus available in our lab, which has a 532 source (mentioned in line 184) does not detect lipids at the concentrations found in Red Stone, while the Raman of Sherlock has a deep UV 248.6-nm laser.

Also, here it must be taken in consideration that the spot size of different Raman instruments also varies wildly. In the case of our Raman, its spot size was 42 μM, while Sherlock spot size is of 50 μM, thus they compare well in this sense at least.

Thanks to your comment, we now comment in lines 194-196 of the revised version of the manuscript that: “...unveiling the critical proper choice of Raman parameters such as laser source and spot size in the detection of different types of organics when concentrations are extremely low”.

- Line 238. This emphasizes that analysis of analog field samples with flight instruments provide important information about the meaning of their results on Martian samples. If these instruments can't find organics in terrestrial samples collected near the ocean, why should they be expected to find them on Mars from rocks laid down billions of years ago?

You are absolutely right!, and this is why we now have included part of your comment in this paragraph as follows (lines 242- 248 of the revised manuscript): “However, when these samples were derivatized with MTBSTFA-DMF (direct derivatization and extraction prior to derivatization), a few peaks were identified, but only in evaporite samples; aliphatic carboxylic acid species derivatized through silylation of their labile group –OH (fig. S19). These results not only unveiled that most Red Stone samples contained organic levels below MOMA detection limits, but also, the critical importance of analog field samples such as Red Stone to validate the next generation of flight instruments.”

- Line 244. SOLID detected signatures of cyanobacteria but previously it was stated that signatures of cyanobacteria were not observed in DNA analysis or in the organic analysis (lines 160-176) and that this implied that cyanobacteria signatures must not be modern.

Does this imply that what SOLID is seeing is antigens of cyanobacteria laid down in the ancient sediments? If so, that would be a very important conclusion.

You are here again right on the spot. Although we do comment that “...consistent with other biosignatures coming from the extant and relict microorganisms...”, we have now improved this paragraph to include your suggestion, adding in lines 256-258 of the revised manuscript; “The detection of ancient remnants of cyanobacteria is particularly interesting in the case of SOLID, as it confirms that the Red Stone river delta had enough water to support photosynthesis millions of years ago, but not anymore as it the Atacama got drier in time (similar to the case of Mars)”.

Reviewer #2 (Remarks to the Author):

This paper highlights the microbial community inhabiting soil samples collected from Red Stone, Atacama Desert and its environmental conditions using the combination of multidisciplinary techniques. Soil parameters of Red Stones were also determined and compared with those detected by ground-based instruments. It is an advanced attempt that provides thorough data for the Mars mission where each technique and parameter were selected and conducted rigorously that will give a good impact in the science field.

The introduction of 'dark microbiome' term is up to date and relevant that can serve as the extension of 'microbial dark matter'. This is the major contribution by the authors which is in line with the advancement of NGS platforms and bioinformatic analysis approaches that contribute towards more explicit exploration in microbial taxonomy and microbial ecology field.

Our responses to specific comments on the manuscript by Reviewer #2

- From the methodology provided, authors mentioned the use of 4 different types of primers in NGS namely 16S, 18S, archaea and cyanobacteria but only 16S results were presented. Please provide the data obtained from other primers.

You are right on this, we only showed NGS 16S data. Although we tried to amplify archaeal DNA, and the amplification with archaea-specific primers was successful (both 1st and 2nd PCRs previous to Illumina sequencing process), sequencing and posterior bioinformatic analyses unveiled that the few resulting sequences obtained above the established quality and length thresholds corresponded in fact to Bacteria.

We think that this could be due to main reasons: the inexistence of Archaea (most likely this reason, as with this very same technique and primers we have unambiguously detected Archaea in other sites of the Atacama (see for example Azua-Bustos et al., 2020, Scientific Reports 10: 19183), or alternatively, an extremely low archaeal biomass. Any of these alternatives, along the variable nature of the primers (necessary to amplify as many Archaea phyla as possible), would result on that the only amplified sequences came from bacterial species, with enough 16S rRNA gene sequence similarity to the employed primers. Thus, we discarded the "archaeal results" in order to make our study as rigorous as possible.

As per 18S data, we found only eight good quality fungal sequences; three usual contaminants found almost everywhere in the world (*Penicillium*, *Aspergillus* and *Rhodotorula*), three known plant pathogens (*Preussia*, *Parastagonospora*, *Jattea*), a human pathogen (*Malassezia*), and a sequence whose genus was not clear in the available databases. Thus, again, to be rigorous, we didn't include 18S NGS data in our report.

Thus, in order to include your pertinent comment on this, we have added in the corresponding Illumina NGS-based 16S and 18S rRNA Sequencing section of the Supplementary Online Materials that "*We note here that no archaea or eukaryotic sequences (except a few well-known fungal contaminants) were found, thus only 16S data is presented in the main report*".

- It is suggested that more analysis can be done on the data of different type of microbes and comparison can be made between them. Multivariate analysis can be conducted to observe the relationship between microbiome data with environmental factors.

You are absolutely right. Thank for this suggestion. We now include the correlation between NGS microbial diversity with both temperature and relative humidity in Figure S7, commenting in line 132 of the revised manuscript that "*A correlation between the highest RH*

values and NGS microbial diversity confirmed the high number of OTUs at the uppermost conglomerates (fig. S7A), also unveiling a slightly higher number of OTUs at the zone where evaporites are located, in line with the role of these minerals in providing a source of water for microbial life due to water vapor absorption in response to changes in RH. Upper Zone OTUs are also located at the zone where the highest surface temperatures are recorded during the day time hours, suggesting that such species are at least thermotolerant.”

- Performing culture dependent technique provide a clearer view of the microbes in Red Stone as extended future studies can be carried out to discover their taxonomic classification and potential in various field. However, data revealed that only few species were isolated: Are those the only 26 isolates obtained in this study?

Yes, only 26 (in fact 25, after the phylogenetic analysis as suggested) unique non-redundant isolates were detected. We note here that these are the usual numbers found in the extremely dry, saline and UV radiated soils of the Atacama, the driest and oldest desert on Earth.

- Or there are more isolates but have not been identified based on 16SrRNA gene sequences? How was the colonies selected?

No, all isolates found were reported. We did not select any colonies, we subculture every colony that grew on the media used, subcultured them, and identified them.

- Please amend the methodology section so that it can be reproduced. This can be done by providing more details on isolation technique (incubation temperature, duration, how the colonies where selected from isolation plates, dilution factor and so on)

This is a very pertinent suggestion. We now expanded the methodology on the section Isolation of soil microorganisms of the supplementary Online Materials file this to include all the information requested.

- Conduct phylogenetic analysis on the 16S sequence of each isolates to get the accurate result of their evolutionary relationship and the taxonomic group that they belong to. BLAST result only provide indication of the closest hit from their database.

You are absolutely right!, we now show the phylogenetic species of all species in Figure S8. Thanks to your suggestion we found that the fungal isolate found in U2 samples did not belong to *Phoma*, but instead was closer to the genus *Ophiosphaerella*. Note that in this case we did not assigned a specific genus to this particular species as we detected that the Phaeosphaeriaceae is quite polyphyletic. Also thanks to your suggestion, we also detected that the bacterial isolate previously assigned as *Labdella* in Upper Zone samples is most likely a new isolate of the genus *Blastococcus*. All of the remaining isolates were confirmed to belong to the genera originally assigned.

Suggestions:

- A more optimized selective isolation technique to be applied in future as commented in the manuscript.

Thanks. We believe that by incorporating the previously requested information we have achieved this. Also consider that we have successfully used the exact same isolation protocol many times before, i. e.:

- <https://www.nature.com/articles/s41598-020-76302-z>
- <https://www.nature.com/articles/s41598-019-47394-z>
- <https://www.nature.com/articles/s41598-018-35051-w>

- Apart from those data, the number of microbes in each soil samples can be estimated using cfu/g.

You are right! We also did this, but did not show it in the first version. Now this information can be found in Table S5.

- As the term 'dark microbiome' is used in the title that shows the importance of this part of study, it is vital for authors to utilize the data obtained and conduct more comprehensive analysis on them to improve this manuscript.

We are really happy to see that you perceive and share with us the great relevance of the “dark microbiome” concept. However, we have decided to perform a more comprehensive and detailed analyses of the dark microbiota in a second report, for which we are already getting new data, and which will focus exclusively on this. Please note that the original intention of this report was to present a new Martian analog model in which the critical issues on detecting biosignatures on Mars could be better highlighted. The finding of the dark microbiome was a happy finding, and we are extremely excited to be already continuing our investigations on the subject. You may also consider that this report, as it is, already has used a wide range of analytical techniques which resulted on 27 figures and tables, and an Online Supplementary File which is 59 pages long.

Thus, we hope that you will concur that the finding of the dark microbiome in Red Stone deserves its own separate report, not only for scientific reasons, but practical as well. In any case, this manuscript includes a first hypothesis explaining the concept, as can be seen in line 149-153 of the manuscript: “ *The Red Stone dark microbiome may be composed by truly novel extant species not found anywhere else on Earth, but it may also be the case that such dark microbiome in fact represents the relict community of microbial species which used to inhabit the Red Stone delta in the distant past, of which no extant relatives are to be found in the existing sequence databases.*”.

- The other experiment such as SYBR green and CARD-FISH analyses contribute to the information of microbial life detected in the soil samples. In addition, the outcome of mineral and environmental data is making this study complete. It would be good if some data can be combined in one or more analysis (multivariate, biological modelling etc) to produce more absolute outcome of this multidisciplinary study.

We do agree that much more can be done! But at some point we also must produce a finished product to provide the basis on the many studies that will follow. We hope that here you agree with us again on that this report is but the foundational stone to more elaborate studies. Also, you may consider that we have, thanks to your pertinent comments, added the initial correlations between environmental parameters and microbial diversity, unveiled the phylogeny of the isolates obtained, and added the information on CFU, which, as said before, resulted in an already quite extensive study.

Reviewer #3 (Remarks to the Author):

I was asked to review the organic geochemistry aspects of this manuscript which I have done. On the organic geochemical side I found that it lacked detail and with the lack of chromatograms from the standard GC-MS analysis makes it difficult to assess. I felt that detailed organic geochemical descriptions are needed in the Supp Data section so this can be used as a benchmark for the SAM and MOMA-like data that is produced.

General Comments

- I was not convinced that organics were detected by Raman, NIR or MIR this needs to be addressed.

Thanks for this comment. We hope that the detailed responses to your comments below have adequately addressed this point.

- Peaks are mislabelled or misidentified in S14

You are absolutely right! We indeed realized that the peak identification of the alkanes was shifted by mistake. This was corrected in the revised figure as detailed in our response on this later.

- All figures need improving with legends and peak labels etc. On the organic geochemistry part of the manuscript I do not feel that it is up to the required standard currently for this journal.

Thanks for this comment. As can be checked below, we have adequately responded and changed everything you suggested, which have much improved legends and peak labels.

Specific points/comments

- Line 159: GCMS should be GC-MS.

You are right, this has been changed as suggested, in line 165 of the revised manuscript.

Line 164: Iso fatty acids are common in numerous bacteria not just sulfate reducers

The reviewer is absolutely right that iso/anteiso fatty acids may come from other types of bacteria, not only sulfate-reducing bacteria. That been said, the latter are known to produce an abundance of these type of branched fatty acids (Taylor and Parkes, 1983, Journal of General Microbiology, 129, 3303-3309), in particular iso- and anteiso-C15 and C17, which have been reported in pure cultures of *Desulfovibrio desulfuricans* (Boon et al., 1977; J. Bacteriol. 129, 1183-1191), enrichment cultures of sulphate-reducing bacteria (Grimalt et al., 1992; Org. Geochem. 26, 509-530), and cultures of Bacillus species (Kaneda, 1967; J. Bacteriol. 93, 894-903).

Nevertheless, we have edited the mentioned sentence to take your comment into account, which now says (line 170 of the revised manuscript): *"The presence of fatty acids with a methyl group at the penultimate position of the acid chain (i.e., iso-17:0) suggest bacterial inputs where sulfate-reducing bacteria may represent a substantial fraction"*

Line 168: How about other common cyanobacteria markers such as hopenes?

You are right again on that cyanobacteria do produce compounds such as hopenes, but we did not detect such biosignatures in Red Stone samples.

Line 173: Which hopane isomers/epimers some are produced by bacteria. Were hopenes detected such as diploptene which is common in cyanobacteria.

Thank you for this very interesting comment! No hopenes (including diploptene) were detected in any of the samples. Only a number of hopanes from 29 to 31 of carbons (C29 $\alpha\beta$, C30 $\alpha\beta$, C31 $\alpha\beta$ S and C31 $\alpha\beta$ R) were found in the mudstone samples, which likely come from prokaryotes, including cyanobacteria, although triterpenoids with a hopane skeleton may occur also in some plants (Kaiser and Arz, (2016), 122: 102–119 and references therein).

Line 177-178: Also dependent on the source not just a measure of metabolic activity

Again, you have a very valid comment on this. We do recognize that the terminology used above (metabolic activity) may not be the most correct. Thus, we have now changed the sentence to say (line 184 of the revised manuscript): *“The ratio of the sum of n-fatty acids over the sum of n-alkanes may provide a measure of the relative extent of freshness of the organics in a given environment”*, since the relative enrichment of functional (carboxylic) groups over plain alkanes may be due to presence of biomass from either active (living biomass) or from recent/subrecent (already dead but geochemically “fresh” biomass) metabolisms.

The use of this ratio as an estimation of the relative “freshness” of the organic matter in a given transect or set of samples is a simplified model. Of course, the specific abundance of one or another individual fatty acid will depend on the source. However, it is the proportion of unsaturated and branched fatty acids that is more dependent on specific biological sources, whereas the linear and straight (normal) fatty acids are quite ubiquitous and abundant in all organisms’ cells. So, by considering only the n-fatty acids in the mentioned ratio of n-fatty acids/n-alkanes we can roughly estimate the relative abundance of fresh organic matter, regardless of its specific biological source. This approach has been proven to be valid by us using other samples of the Atacama Desert:

- <https://www.nature.com/articles/s41598-020-76302-z>

- <https://www.nature.com/articles/s41598-018-35051-w>

Line 179: Depends on the diagenetic conditions need to be anoxic otherwise the functionalised components just get further oxidised.

You are right again on that the degradation of the fatty acids to more oxidized or reduced compounds will depend on the environmental redox conditions, among other factors. Still, pure C-H skeletons (alkanes) are the last residue chains remaining after degradation of more labile organic chains, such as fatty acids. So, regardless of how fatty acids degrade (depending on the depositional environment), a relative enrichment of n-alkanes over functional groups is expected as organic matter matures (i.e., over time). Accordingly, the proportion of n-fatty acids over n-alkanes do provides a measure of the “relative freshness” of organic matter along a temporal/spatial transect in a given environment.

Lines 181-3: Over simplification as stated above could be down to different source and differing diagenetic/redox conditions

Please see detailed responses above. Also, to take your comment on this in account, we have changed this phrase to (line 188 of the revised manuscript): *“Samples from the upper part of the stratigraphy and evaporites showed the highest ratios (≥ 16 ; Table 1), suggesting that*

these samples contain the fresher/more recent biomass, consistent with the highest microbial diversity determined by NGS and the highest availability of water in these samples”.

Line 184: What is the detection limit of Raman (it is generally an insensitive technique for organic detection) is it not well below the already stated TOC?

This is indeed a very pertinent question. The limit of detection has been discussed in some papers focused on Astrobiology because it has implications on the analytical protocols and spectrometer design (see for example Vandenabeele et al. 2012). It is well known that the detection of a specific Raman band signature may be dependent upon several measurement aspects, including laser power, the sample illumination, the observation geometry and the sensitivity of the detector.

Thus, thanks to your comment we now comment on line 194 of the revised manuscript: *“...unveiling the critical proper choice of Raman parameters such as laser source and spot size in the detection of different types of organics when concentrations are extremely low”.*

The limit of detection also depends on how active is each compound in Raman: pigments are strongly active in contrast to aminoacids, which are weaker Raman scatters. Thus, each compound would have an optimal signal according to these parameters. Most of the minerals usually have good signal with a green laser at 532 nm, while organics, which usually produce fluorescence, are better detected with lasers in wavelengths out of the visible range.

In addition, Raman intensities are relative, which means that if there are minerals with high signal organics will be hard to detect. This is discussed in one of our most recent reports (<https://www.nature.com/articles/s41598-022-09684-x>), in which we used the same instrument than in this manuscript to analyze highly altered samples of difficult analysis by Raman, also comparing the results with the RLS instrument.

As for the detection limit, we showed in another report (<https://www.liebertpub.com/doi/abs/10.1089/ast.2019.2100>) that the detection of organics with Raman were easier if they were in mineral matrices with poor or no signal, like halite. For example, the limit of detection of our instrument for lipids was 0.5 mg g⁻¹ (0.05 %dw).

Thus, our instrument would be sensitive enough, or at least, at the limit of detection to see Raman active organics associated to minerals, but it is less efficient with Red Stone samples. Add to this the “un-cooperating” mineral background, to explain why this specific Raman could not see any organics, which is the precise point we wanted to make for the case of Martian samples.

Line 201: What is the detection limit for organics by DRIFTS is it not above the already stated levels. Not convinced by the NIR or MIR traces that there are any peaks that can be unambiguously identified as being due to C-H stretching/overtone. NIR main C-H overtone is at 1700 nm. The main C-H peaks in MIR should be ~2900 cm⁻¹ but not highlighted if present. Approx MIR detection limits for orgC are ~0.1% under ideal conditions which these are not (e.g. DTGS detector etc.)

You again have a valid point here. Unfortunately, in the case of Red Stone samples, the fundamental C-H stretching vibrations were covered by the stronger calcite bands which occur right there, and we couldn't claim detection of these bands.

However, the band around 7340 cm⁻¹ can be attributed to a second overtone of C-H stretching or a combination band involving C-H stretching vibrations and other vibrational modes, as previously reported for a number of lipids:

- Bista, R. K., & Bruch, R. F. 2008, Near-infrared spectroscopy of newly developed PEGylated lipids. *Spectrochimica Acta Part A: Molecular and Biomolecular Spectroscopy*, 71:410-416
- Ismail, A. A., Nicodemo, A., Sedman, J., Van de Voort, F., & Holzbaur, I. E. 1999. *Infrared spectroscopy of lipids: principles and applications* (pp. 235-269). CRC Press: Boca Raton, FL.
- Daoud, S., Bou-Maroun, E., Waschatko, G., Horemans, B., Mestdagh, R., Billecke, N., & Cayot, P. 2020. Detection of Lipid Oxidation in Infant Formulas: Application of Infrared Spectroscopy to Complex Food Systems. *Foods*, 9(10):1432) and also alkanes:
- Clark, R. N., Curchin, J. M., Hoefen, T. M., & Swayze, G. A. 2009. Reflectance spectroscopy of organic compounds: 1. Alkanes. *Journal of Geophysical Research: Planets*, 114-E3).

Limits of detection are tricky to be determined in Infrared Spectroscopy, and even using a DTGS detector we are usually able to detect weak organic bands in the MIR when organics are at very low concentration overall. That's because organics are usually not homogeneously distributed in the samples, rather they are localized/concentrated in small spots, and if you hit those hotspots you can detect them. You can find proofs of this assertion in the following papers:

- UV Irradiation and Near Infrared Characterization of Laboratory Mars Soil Analog Samples. T. Fornaro, J. Brucato, G. Poggiali, M. A. Corazzi, M. Biczysko, M. Jaber, D. I. Foustoukos, R. M. Hazen, A. Steele. *Front. Astron. Space Sci.* 2020
- UV Irradiation of Biomarkers Adsorbed on Minerals under Martian-like Conditions: Hints for Life Detection on Mars. T. Fornaro, A. Boosman, J. R. Brucato, I. L. ten Kate, S. Siljeström, G. Poggiali, A. Steele, R. M. Hazen, *Icarus* 2018, 313, 38-60.

Figure S11: X-axis needs to be reversed and it is also usual to plot the Y-axis as log(1/R) instead of reflectance

You are absolutely right on this, a new figure (now Figure S13) has been changed in accord to your request.

Figure S14 has mis-label/identified compounds; C16 fatty acid would not elute next to the C16 alkane but just before n-C20. Why is only the m/z 57+71 mass chromatogram shown and not the TIC with the compounds mentioned in the main text (line 216). Peak labels need to be described in the legend

You are right on this again. The figure actually did not come out how it was supposed to in the paper: the grey lines have disappeared from the figure when it was transferred to the main manuscript. We have corrected this in the revised figure (now figure S16).

In addition, we processed the alkanes peaks with a second thorough analysis and there was indeed a shift in the retention times of the alkanes. The alkanes were primarily identified with their MS as well as their retention times using an alkanes internal standard which was run in the same GC conditions as the experiment for comparison. However, we realized that C10

and C12 alkanes were missing in the chromatogram of the C10-C40 alkanes standard (likely because they were too volatile) which made us shift the peak identifications. Thank you for carefully reviewing that figure and help us correct that mistake! We confirm that C16 fatty acid is eluting at 36.96 min, right before C20 which is eluting at 37.47 min. All the peak identifications were revised on the figure to correct that mistake and the text in the paper was corrected to clarify the detection of C12-C35 alkanes (instead of C8-C31).

We chose to show the m/z 57 + 71 mass chromatogram to extract the major ion mass fragments of the alkanes from the TIC which better highlighted the peaks compared to the TIC only. The alkanes were present in low abundance and were barely visible in the TIC chromatogram.

In addition, the peak labels are now described in the legend as requested by you.

Line 214: No mention of the alkenes if pyrolysis is occurring the alkenes should be evident, if they are not is thermal desorption a more dominant process due to the relatively slow heating rate? No mention of how the mineral assemblage would affect the products seen in the pyrolysate, which can have a significant affect. Was no contamination detected (all samples are contaminated just to different extents)? Were any standards run to check the pyrolysis system was working correctly?

Thank you for this remark. Alkanes/alkenes doublets are usually associated with the pyrolysis-GCMS of terrestrial biological kerogens and related to the decarboxylation of fatty acids. Fatty acids were indeed detected using SAM-like pyrolysis (specifically C16 and C18) and SAM-like derivatization.

However, only alkanes were detected in our SAM-like pyrolysis experiments. When alkanes/alkenes pairs are detected, the alkenes are usually present in lower abundance compared to the alkanes. The relative abundance of the alkanes detected in our experiments was already very low compared to other molecules: they were barely detectable in the TIC and their detection was possible and confirmed by extracting the ion chromatogram of m/z 57 + m/z 71 (major mass fragments of the alkanes). Therefore, it is possible that the relative abundance of the alkenes was too low to be detectable in these experiments. Another possible explanation is that they may have reacted with H₂ present in the system and that the alkenes were transformed and saturated into alkanes.

Blanks were performed before and between each pyrolysis-GCMS analysis and no contamination was detected. We ran a C10-C40 alkanes standard to confirm the identification of the alkanes in our analyses.

Figure S15: Are the peak labels alkanes or fatty acids? If they are fatty acids surely C16 should be the largest peak? The trace should bare some relationship to the straight pyrolysis runs which it does not appear to do. How will the minerals affect the derivatisation, release of bound water (as can be seen in S16) will both damper the reaction and can make the derivatised products unstable.

The peak labels are indeed fatty acids. As for figure S14 (now Figure S17), the grey lines highlighting the fatty acids in each chromatogram disappeared when the figure was transferred to the main manuscript.

So, you are correct, C16 fatty acid were the most abundant molecule closely followed by the C18 fatty acid, which is consistent with the SAM-like pyrolysis results (no derivatization), as already stated in the main text.

It did not appear like that on the figure, because of a conversion issue of the figure which removed the grey squares highlighting the actual fatty acid peaks (see screenshot of the good figure below). The main peaks that seemed to be C12 and C14 on the figure from the paper that was originally submitted are actually MTBSTFA byproducts: by-silylated water at 24.16 min and another unidentified MTBSTFA byproduct at 29.05 min (including major ion mass fragments m/z 73, 75, 147). This has now been corrected, and we added a label on the figure to clarify this and explained in the legend that these peaks correspond to MTBSTFA byproducts.

The mineralogy of the sample (e.g., presence of oxides, release of large amounts of water from clay minerals) can indeed impact the derivatization. The presence of mono-silylated (not shown in the chromatogram) and bi-silylated water (24.16 min) in the chromatogram indicates side reaction with water. However, although these peaks are present in relative high abundance, they are not saturated and did not prevent the derivatization reaction to occur. This is confirmed by the really good detection of the 3-fluorovaline internal standard at 20.32 min which is a standard (see the new S17 figure) which is a standard used specifically to ensure that derivatization worked.

Line 232: MOMA heating at 400 C is below pyrolysis temperatures and is thermal extraction

Thanks for this pertinent comment. The term pyrolysis in this specific case is correct, since at 400°C, breakage and recombination of compounds is often observed, particularly when these are adsorbed on minerals, as is the case of Red Stone samples. It is therefore not simply a matter of thermal extraction. We have recently reported this specific issue in: Royle, S. H., et al. (2021). Pyrolysis of carboxylic acids in the presence of iron oxides: implications for life detection on missions to Mars. *Astrobiology*, 21(6), 673-691.

Table 1: The units are $\mu\text{g g}^{-1}$ dry weight as not as written in legend. No mention of where the values are derived for the concentrations

We apologize but don't understand exactly what the review meant by "not written in legend", as in fact the units of lipids are indicated in the table caption.

Lipid concentration was derived by integration of the peak areas and conversion to $\mu\text{g g}^{-1}$ by applying the response factor of deuterated standards (tetracosane-D50 and myristic acid-D27) added to both the sample (before extraction) and an external calibration curve (i.e., mix of reference n-alkanes C10-C40 and n-fatty acids C8-C24, depending on the polarity fraction). We haven't mentioned this as this is the usual way of calculating concentrations from chromatograms peaks.

That been said, in order to take your comment in account, a new sentence to better explain this has now been included in the Table caption and the corresponding Supplementary Online Materials section.

Figure S17: Why are the products not labelled?

Thank you for this very pertinent comment. The legend of this figure (now Figure S19) read: *“Portion of the chromatogram obtained after direct derivatization with MTBSTFA-DMF reagent of the evaporites sample. The peaks marked with asterisks correspond to derivatized organic compounds. The other peaks are either byproducts of the derivatization process or column artifacts”*. In this figure the labels of the derivatized product were not specified as we were unsure of their exact identification.

Thus, in order to take your comment in account, Fig. S17 (now S19) legend now states *“Portion of the chromatogram obtained after direct derivatization with MTBSTFA-DMF reagent of the evaporites sample. The peaks marked with asterisks correspond to derivatized organic compounds (attributed to aliphatic carboxylic acids but no strict identification could be found). The other peaks are either byproducts of the derivatization process or column artifacts”*.

Supp data

Biosignature analysis (GC-MS): Where are the chromatograms for the more traditional wet chemistry organic geochemical techniques. The recoveries are rather low

You are right. All the chromatograms obtained for the apolar and acidic fractions of the Red Stone samples are now included in the Supplementary Online Materials section.

The recoveries obtained here for the internal standards ($71 \pm 14\%$) are in the range of others obtained in similar studies ($69 \pm 18\%$ to $84 \pm 25\%$) by our and other research groups:

- Megevand et al. (2022). *Frontiers in Microbiology* 13:811904. doi: 10.3389/fmicb.2022.811904.
- D. Carrizo et al. (2022). *Astrobiology*, DOI: 10.1089/ast.2021.0036.
- Lezcano et al. (2022). *Frontiers in Microbiology*, 13, 799360. doi: 10.3389/fmicb.2022.799360.
- Vega-García et al. (2021) *Science of the Total Environment* 755, 142662. doi.org/10.1016/j.scitotenv.2020.142662.
- Azua-Bustos et al. (2020). *Scientific Reports* 10(1),19183.
- Sánchez-García et al. (2020). *Scientific Reports* 10(1), 21196.
- Carrizo et al. (2019). *Astrobiology* 19, DOI: 10.1089/ast.2018.1963.
- Sánchez-García et al. (2019). *Frontiers in Microbiology* 9, doi: 10.3389/fmicb.2018.03350.
- Vonk, Sánchez-García et al. (2012). *Nature*, 489, 137-140. doi: 10.1038/nature11392.
- Van Dongen et al. (2008). *Global Biogeochemical Cycles*, VOL. 22, GB1011, doi:10.1029/2007GB002974, 2008.

Reviewer #1 (Remarks to the Author):

I applaud the authors embracing and implementing the changes requested by the reviewers. I have flagged a small number of typo issues below. Other than these, I believe the paper to be ready for publication.

Line 256 a typo in the word " interestiSng " it should be "interesting"

Line 258 , delete "it" in the phrase "but not anymore as it the Atacama got drier in time"

Page 42 of the supplemental data section. The sentence (top of page and appears to be part of the figure caption for Figure S20) is out of place.

"Raw lipid chromatograms in the acidic (Fig. C1-C8) and apolar (Fig. C9-C17) lipid fractions of the Red Stone." I think it belongs in the following paragraph under Total Ion Chromatograms.

Reviewer #2 (Remarks to the Author):

This manuscript consist of valuable data on the insights of environmental factor and microbial life inhabiting Red Stone in Atacama Desert using various muldiciplinary approaches that will contribute towards the attempt to unravel the life in Mars. Manuscript is well written and amendment had been made that make it look explicit. Thank you for your response on my comments, really looking forward for more reports from your group. Just some comments from me:

1. Well done on Figure S8. Please refer to supplementary material file for suggestion on the legend for this figure.
2. Table S5 – i) this table is not being mentioned in any part of the manuscript including the article file and supplementary materials. ii) CFU/g should be presented in one decimal place. Please do correction on the data and include them in the manuscript/main text if reliable

In their review of the second version of this manuscript, reviewer #2 added some comments to the manuscript file. These comments were forwarded to the authors, who replied as included in this Peer Review File

Reviewer #3 (Remarks to the Author):

I am satisfied that all my comments from the original submitted manuscript have been addressed by the authors and I would recommend the manuscript for publication.